# FedDSE: Distribution-aware Sub-model Extraction for Federated Learning over Resource-constrained Devices

Submission Id: 566

## ABSTRACT

Sub-model extraction based federated learning has emerged as a popular strategy for training models on resource-constrained devices. However, existing methods treat all clients equally and extract sub-models using predetermined rules, which disregard the statistical heterogeneity across clients and may lead to fierce competition among them. Specifically, this paper identifies that when making predictions, different clients tend to activate different neurons of the entire model related to their respective distributions. If highly activated neurons from some clients with one distribution are incorporated into the sub-model allocated to other clients with different distributions, they will be forced to fit the new distributions, which can hinder their activation over the previous clients and result in a performance reduction. Motivated by this finding, we propose a novel method called FedDSE, which can reduce the conflicts among clients by extracting sub-models based on the data distribution of each client. The core idea of FedDSE is to empower each client to adaptively extract neurons from the entire model based on their activation over the local dataset. We theoretically show that FedDSE can achieve an improved classification score and convergence over general neural networks with the ReLU activation function. Experimental results on various datasets and models show that FedDSE outperforms all state-of-the-art baselines.

## CCS CONCEPTS

• **Computer systems organization** → **Embedded systems**; *Redundancy*; Robotics; • **Networks** → Network reliability.

## KEYWORDS

Federated Learning; Submodel extraction; Distribution-aware

**ACM Reference Format:**
Anonymous Author(s). 2018. FedDSE: Distribution-aware Sub-model Extraction for Federated Learning over Resource-constrained Devices. In *Proceedings of Make sure to enter the correct conference title from your rights confirmation emai (Conference acronym 'XX).* ACM, New York, NY, USA, 25 pages. https://doi.org/XXXXXXX.XXXXXXX

## 1 INTRODUCTION

With the proliferation of edge devices like IoT and sensors, huge amounts of data are generated continuously, which can be used to train efficient machine learning models. However, the raised privacy concerns make it difficult to collect big data from edge devices and send them to a central cloud for training. Federated Learning (FL) [22, 32], which enables clients to collaboratively train machine learning models in a decentralized manner without revealing their private raw data, is an emerging paradigm that has been adopted in various fields including medical image processing [44] and recommendation systems [11]. However, to deploy FL in practical edge environments, it is necessary for the resulting systems to not only preserve the privacy, but also satisfy the common pragmatic constraint, i.e., constrained resources such as energy, computation, communication, and memory of edge devices [4, 14, 25].

To address the aforementioned issues, extracting the sub-model from the entire model appears to be an effective solution, which is also called partial federated learning, where each device only trains a sub-model of the full global model. Two categories of sub-model extraction methods for FL have been proposed: parameter sparsifying methods [3, 19, 26, 37] and neuron pruning methods [2, 6, 9, 16]. Parameters sparsifying methods extract sub-models by selecting specific parameters from the entire neural network based on the lottery ticket hypothesis [13]. Although they effectively reduce the computation and communication costs, recent works [4] have shown that such methods do not reduce the memory trace because the activation outputs from neurons are much larger than the original parameters. Neurons pruning methods [2, 6, 9, 16] extract sub-models by selecting a subset of neurons from the entire neural network. For example, FedRolex [2] selects neurons in a rolling way for each client. Considering their great advances in terms of memory efficiency, this paper mainly focuses on the category of neuron pruning methods.

Although current neuron pruning methods are effective in reducing memory usage, they do not account for statistical heterogeneity (i.e., non-identically distributed data) [21, 28, 29, 33], potentially leading to decreased performance. Specifically, this study reveals the competition between clients with different data distributions when only sub-models are locally trained. We observe that clients tend to activate different neurons within the model during prediction, closely linked to their respective data distributions. As data distribution is neglected, the neurons highly activated for clients with one distribution may be extracted into a sub-model designated to other clients with distinct distributions. Newly-assigned clients may find it challenging to obtain effective representations over local datasets via the sub-model with limited capacity, as they have to force the neurons strongly linked to previous clients to adapt to these new distributions. On the other side, such a re-fit process may also in turn hinder the activation of these neurons over the previous clients and result in a performance reduction.

Motivated by this finding, we propose a simple yet effective method FedDSE to reduce the conflicts among clients by extracting

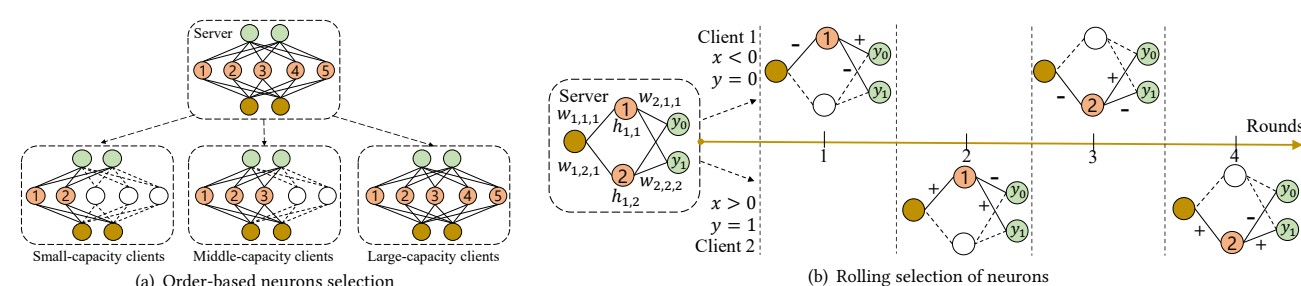

(a) Order-based neurons selection

(b) Rolling selection of neurons

**Figure 1: Illustration of existing methods that extract neurons with pre-defined rules. (a) An example of three types of clients with order-based neurons selection (Fjord [16] and Hetero [9]). Neurons 4 and 5 may only be trained a few times due to the limited number of large-capacity clients. (b) An example of two clients (different rows) selecting neurons in a rolling way (FedRolex [2]). Clients may compete for neurons to fit their respective distinct distributions.**

sub-models based on the data distribution of each client. The main idea of FedDSE is to empower each client to adaptively extract neurons from the entire model based on their activation over the local dataset, where neurons with the largest magnitude are selected. In this way, the conflicts can be minimized since every client is assigned its most appropriate neurons instead of the ones activated for other clients with different distributions. Experiment results on different datasets and models show that FedDSE can significantly improve the training efficiency under the constraint of limited memory compared to baselines. Our contributions are:

- To the best of our knowledge, this paper is the *first* to consider statistical heterogeneity in FL with sub-model extraction. Our findings reveal that clients with distinct distributions tend to activate different neurons, leading to conflicts among them when the neurons are not assigned properly.
- We propose a novel training method, FedDSE, to extract sub-models for each client based on their data distributions. In FedDSE, the neurons of the sub-model are chosen based on their levels of activation over the local dataset of each client, enabling us to assign the most appropriate neurons to each client.
- We establish a theory for the convergence of FedDSE on general neural networks with ReLU activation function, which shows that our method has an asymptotic convergence rate.
- To validate the efficiency of the proposed method, we compare FedDSE with state-of-the-art methods. Evaluation results show that FedDSE can improve the performance by up to 2.72%.

## 2 RELATED WORKS

Many approaches have been proposed to realize FL over memory-limited devices, which can be categorized into two main types based on whether the weights of the global model are updated.

### 2.1 Training masks from the fixed-weights global model

This category of works initially comes from the centralized scenario, where the masked model of a dense network with random weights performs surprisingly well without ever training the weights [1, 35, 36, 46]. Considering this phenomenon, some recent works seek to find such a mask to reduce the communication budget in FL, while simultaneously compressing the given global dense network [18, 27, 41]. Although these methods achieve success separately,

their targets are totally different from ours. For example, Li et al. [27] focus on the personalization of local models over different clients via various masks. Anish et al. [41] and Isik et al. [18] seek to reduce the computation and communication costs via the 1-bit mask. In contrast, this paper mainly focuses on the issue of limited on-device memory. While these prior methods can also reduce the memory usage by reducing the size of parameters, they cannot reduce the size of activation which consumes much more memory [4]. Besides, these methods rely on a dense network, which may also potentially increase the memory usage.

### 2.2 Training sub-model weights extracted from the global model

These methods train the global model by updating the weights of the extracted sub-model, which are further classified into two categories, i.e., parameter sparsifying methods and neuron sparsifying methods. Parameters sparsifying methods extract sub-models by selecting specific parameters from the entire neural network [3, 19, 26, 37] , which are usually based on the theory of the lottery ticket hypothesis [13]. Although they effectively reduce the computation and communication costs, recent works [4] have shown that such methods do not reduce the memory trace because the activation outputs from neurons are much larger than the original parameters. Another line of methods is to extract the sub-model by pruning neurons from the global neural network [2, 6, 9, 16, 30]. For example, an earlier method randomly prunes neurons from the global neural network for each client [6]. For the heterogeneous edge devices, Fjord [16] and Hetero [9] employ a similar approach. They manually define a neuron-order before training and construct sub-models for each client based on its memory constraints, and then select neurons in accordance with this pre-defined order. However, ordered extraction requires an adequate number of high-capacity devices to accommodate the complete model. Otherwise, as illustrated in Figure 1(a), many neurons located towards the tail-end of the sequence may not be adequately trained, resulting in degraded performance. In practice, the number of large-capacity devices is generally far less than the low-capacity devices, which restricts its application. Considering this limitation, the recent work FedRolex [2] extracts the sub-model by selecting neurons in a rolling way for each client such that all neurons can be trained equally. However, such a method may cause competition

among clients, as we will illustrate later. These neuron pruning methods are most close to this paper. But different from them, we take the statistical heterogeneity into account when extracting sub-models for different clients.

## 3   PRELIMINARIES

**Basics of deep neural network**. We consider a deep neural network with $L$ layers, and each layer $l$ contains $m_l$ neurons. We denote the weight parameters of the model as $\mathbf{w}$ and the parameters of the $l$-th layer as $\mathbf{w}_l = [w_l, b_l]$ with the weights $w_l$ and bias $b_l$. For each $i$-th neuron in the $l$-th layer, we compute its activation output as $h_{l,i} = \sigma(w_{l,i}\mathbf{h}_{l-1} + b_{l,i})$, where $\sigma(\cdot)$ is the nonlinear activation function (e.g., ReLU), $w_{l,i}$ and $b_{l,i}$ denote the weights/bias for this neuron, and $\mathbf{h}_{l-1}$ represents the outputs of all neurons in the previous layer, i.e., $\mathbf{h}_{l-1} = [h_{l-1,1}, \ldots, h_{l-1,m_{l-1}}]$. For simplicity, we denote all weights of the network as $\mathbf{w} = [\mathbf{w}_1, \ldots, \mathbf{w}_L]$.

**Problem formulation.** Our objective is to allow all clients to collaboratively train a global model via FL. We presume that there are $N$ clients, and each client $n$ has access only to its own private dataset $\mathbb{D}_n := \{x_i^n, y_i\}$, where $x_i$ represents the $i$-th input data sample, and $y_i \in C = \{1, 2, \cdots, C\}$ represents the corresponding label of $x_i$. The number of data samples in dataset $\mathbb{D}_n$ is represented by $D_n$. $\mathbb{D} = \{\mathbb{D}_1, \mathbb{D}_2, \cdots, \mathbb{D}_N\}$, with $N = \sum_{n=1}^N D_n$. The goal is to train a global model $\mathbf{w}$ by minimizing the total empirical loss over the entire dataset $\mathbb{D}$:

$$\min_{\mathbf{w}} F(\mathbf{w}) := \sum_{n=1}^N \frac{D_n}{D} F_n(\mathbf{w}), \text{ where } F_n(\mathbf{w}) = \frac{1}{D_n}\sum_{i=1}^{D_n} f(\mathbf{w}; x_i, y_i),$$
(1)

where $F_n(\mathbf{w})$ denotes the local loss function of the $n$-th client, which measures its private dataset's local empirical risk, and $f(\cdot)$ is the cross-entropy loss function that quantifies the difference between the predicted and ground-truth labels.

## 4   CHALLENGES AND MOTIVATIONS

### 4.1   Resource Properties of Edge Devices

**Limited Memory**. Different from servers in the cloud, edge devices generally have limited capability in terms of memory, energy, communication, and computation. For example, the device Raspberry Pi 1 Model A, which is widely used in edge applications, e.g., smart home [20], only has a memory of 256 MB. Although the memory is sufficient for the inference of neural networks, e.g., the popular ResNet18 where the memory footprint is approximately 60 MB in the inference process, the device can hardly support its training. Specifically, training ResNet18 with a small batch size of 8 requires a memory of 569.67 MB, which far exceeds the memory limit. The available memory will become even less when other applications are running on the device. On the other hand, energy consumption is also strongly related to memory access. Widely used edge devices mobile-phone which are usually equipped with intelligent accelerators [1]. The memory of these mobile phones is composed of DRAM in the CPU and SRAM in the accelerator. Under the 45nm CMOS technology [15], a 32bit off-chip DRAM access consumes 640 pJ, which is two orders of magnitude larger than a 32bit on-chip SRAM access (5 pJ) or a 32bit float multiplication (3.7 pJ). Despite

the energy efficiency of the SRAM, the accelerator usually has limited memory of SRAM. For instance, TPU [17] only has 28MB of SRAM which is even smaller than the training memory footprint of a small network MobileNetV2 using a small batch size of 1 [5]. This leads to numerous resource-intensive DRAM accesses, consequently consuming significant energy and depleting the battery of edge devices. In fact, SSD or Flash access costs even more energy than DRAM. *These properties of memory indicate the necessity of training the sub-model on each local device.*

**Asymmetric network bandwidth of edge devices**. Most current methods use sub-models downloaded from servers to reduce the download bandwidth. However, it is worth noting that upload bandwidth is often much lower than download bandwidth and is the main bottleneck for communication efficiency. This can be seen by summarizing the bandwidth of mobile networks provided by different global telecom operators[2]. In fact, the download bandwidth can be up to 7.7 times larger than the upload bandwidth. Given this, *a natural improvement idea would be to download the full model from the server to improve the training performance while only uploading sub-models to ensure efficient communication.*

### 4.2   Extracting Neurons with Pre-defined Rules May Cause Competition

Here we demonstrate the necessity of extracting client-specific neurons based on their unique data distribution in FL. We present an analysis of the limitations of FedRolex [2], which is currently the state-of-the-art method for FL with sub-model extraction. Specifically, Figure 1(b) illustrates a simple binary classification problem for single-dimension data, where the label $y = 0$ corresponds to data points $x \leq 0$ and $y = 1$ is assigned to $x > 0$. All samples with label $y = 0$ are allocated to the first client and those with label $y = 1$ to the second client. A two-layer neural network with two hidden neurons and ReLU activation function is employed for this classification task. Our example reveals that during training, neurons can become biased towards one particular client and fail to adapt well to other clients' data distribution. For instance, after the first round, neuron 1 is trained to recognize data $x < 0$ of client 1 by updating the parameter $w_{1,1,1}$ to negative (denoted by '-'). In the next round, it is designated to the second client and may struggle to adjust to the new data $x > 0$ by updating the parameter $w_{1,1,1}$ from negative to positive ('+'). On the other side, the adjusting process will also hinder its activation over data from the previous client. Such a conflict is due to the neglect of data distribution when extracting neurons into the sub-model for each client, where the neurons strongly linked to clients with one distribution may be designated to other clients with different distributions. To present this problem formally, we establish the following theory for the general two-layer neural networks.

THEOREM 1. *Consider a two-layer neural network employing the ReLU activation function and being trained with a cross-entropy loss. Let $\mathbb{D}_{n_1}$ comprise samples belonging to class $s$, and $\mathbb{D}_{n_2}$ consist of samples from class $c$, representing the datasets of clients $n_1$ and $n_2$ respectively. Let $h_i(\mathbb{D}_{n_1}) = \sum_{j=1}^D ReLU(\mathbf{w}_i^T\mathbf{x}^j)$ represent the sum of*

---

[1]https://ai.googleblog.com/2019/11/introducing-next-generation-on-device.html

[2]https://www.opensignal.com/reports/2023/02/global-state-of-the-mobile-network-experience-awards

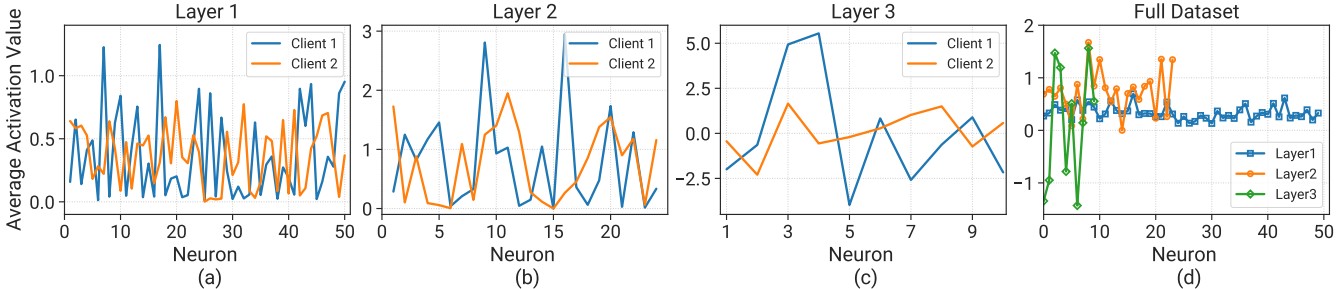

**Figure 2: Comparison of activation distributions of a 3-layer MLP on MNIST. (a-c) Activations of two clients on layer-1, 2 and 3. (d) Activations of different layers trained on the full dataset.**

activations of the $i$-th selected hidden neuron across dataset $\mathbb{D}_{n_1}$, with $D$ denoting the dataset size. Subsequently, training the sub-model $\hat{\mathbf{w}}$ on dataset $\mathbb{D}_{n_2}$ and denoting $p_s^k$ as the probability score of sample $\mathbf{x}^k \in \mathbb{D}_{n_2}$ over the trained sub-model, with a learning rate $\eta > 0$, yields the following observations:

• When the dataset $\mathbb{D}_{n_1}$ of client $n_1$ is homogeneous to the local training dataset $\mathbb{D}_{n_2}$ of client $n_2$, i.e., $\sum_{\mathbf{x}^k \in \mathbb{D}_{n_2}} p_s^k (\mathbf{x}^k)^T \mathbf{x}^j \geq 0$ for each sample $\mathbf{x}^j \in \mathbb{D}_{n_1}$, the activation sum $h_i(\mathbb{D}_{n_1})$ increases, where the augmentation can be as high as $\eta \sum_{\mathbf{x}^j \in \mathbb{D}_{n_1}} \sum_{\mathbf{x}^k \in \mathbb{D}_{n_2}} p_s^k (w_{2,c,i} - w_{2,s,i})(\mathbf{x}^k)^T \mathbf{x}^j$.

• Conversely, when the dataset $\mathbb{D}_{n_2}$ of client $n_1$ is heterogeneous to the local training dataset $\mathbb{D}_{n_2}$ of client $n_2$, i.e., $\sum_{\mathbf{x}^k \in \mathbb{D}_{n_2}} p_s^k (\mathbf{x}^k)^T \mathbf{x}^j \leq 0$ for each sample $\mathbf{x}^j \in \mathbb{D}_{n_1}$, the activation sum $h_i(\mathbb{D}_{n_1})$ decreases, where the reduction is $Min(h_i(\mathbb{D}_{n_1}), -\eta \sum_{\mathbf{x}^j \in \mathbb{D}_{n_1}} \sum_{\mathbf{x}^k \in \mathbb{D}_{n_2}} p_s^k (w_{2,c,i} - w_{2,s,i})(\mathbf{x}^k)^T \mathbf{x}^j)$.

The proof can be found in B. Theorem 1 suggests that clients possessing homogeneous data distributions will mutually amplify their activation learning, while clients with heterogeneous data distributions will mutually diminish each other's activation.

## 4.3 Neuron Properties of DNNs in FL

To investigate the principle of neuron competition, we seek to present the properties of DNN neurons in FL. Through profiling the training process of clients over local datasets, we find neurons are activated differently for specific clients. To demonstrate the potential in extracting neurons, we track the training progress of different layers of a Multilayer Perceptron (MLP) as an example. MLP is a simple and popular model for image classification, consisting of multiple fully-connected layers. Figure 2 compares the activation distributions (i.e., the output feature map produced by a DNN layer) of a three-layer MLP fully trained on the MNIST dataset. The number of neurons for layers 1 to 3 are 50, 24 and 10 respectively. We take the average activation of each neuron over 256 data samples. From Figure 2, we can get the following insights:

• *Each client activates distinct neurons.* Figure 2 (a)-(c) depict the activation values of neurons in different layers for two clients (five clients in total for experiments and we only take two for better illustration here). Obviously, there exists a huge variance between the activation distributions of those two clients. Their curves barely overlap and those neurons with high activation

values also vary for each client. For instance, in layer-2, neuron-16 generates a larger activation value for client-1 while a lower value for client-2, indicating this neuron is activated more by local data of client-1. Similarly, other clients also show their correspondingly stressed neurons in each layer. This pattern reveals a natural strategy: *each client can extract neurons from the global model based on their most activated ones.*

• *The activations of different layers differ.* To further verify the above point, Figure 2(d) shows the average activations of each layer on i.i.d dataset. The values of each layer distinguish much between each other: activation values of the first layer tend to be stable while subsequent layers show more fluctuations. *The activation distributions vary as the model goes deeper, indicating that comparing activations of different layers is insufficient to unmask neuron properties for each client.*

In fact, we have the following proposition to show that the activation magnitude is strongly related to the classification accuracy which is represented as the probability score for each class.

PROPOSITION 2. *Given a well-converged two-layer neural network with the ReLU activation function, high activation values have a large impact on the probability score than low activation values. Specifically, for any sample $\mathbf{x}$ with label $y = c$, the ratio of impact over probability score $p_c$ between a high activation $h_H$ and a low activation $h_L$ is approximately $e^{\alpha(h_H^2 - h_L^2)}$, where $\alpha > 0$ is a constant.*

The proof can be found in B. Proposition 2 shows that higher activation contributes more to the probability score of the classification label. Jointly considering Proposition 2 and Theorem 1, we can intuitively get that the accuracy of the global model over the dataset of some specific client will be reduced when the corresponding neurons with large activation are allocated to other clients of which their data distributions are heterogeneous to this client. More explanations are discussed in Appendix A.

## 5 FEDDSE DESIGN

Motivated by the above findings, we propose to extract a sub-model for each client based on its data distribution, where the detailed workflow is presented in Algorithm 1. Our method FedDSE has the following innovations. First, considering the sufficient download bandwidth, we allow each client $n$ to pull the entire model $\mathbf{w}$ from the server. Second, based on the basic property of neural networks that inference consumes much less memory than training, each client $n$ selects neurons by only running inference over the model

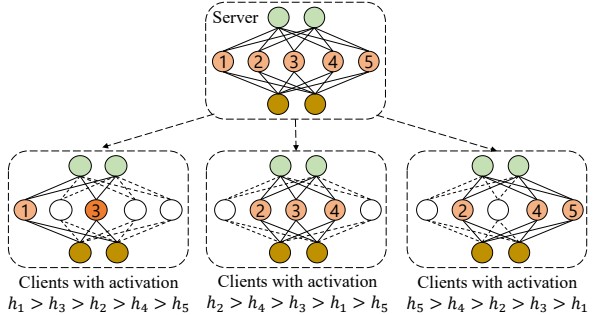

**Figure 3: Clients extract sub-models based on the magnitude of neuron activation.**

with a portion of its local dataset. Third, based on the observation that the magnitude of neuron activation differs a lot for different layers, each client extracts neurons in a layer-by-layer manner, which does not requires caching the activation of previous layers.

Specifically, for each layer $l$, the client $n$ only remains the top ratio $r$ of neurons *in a weighted sampling manner* and prunes the other neurons to obtain the sub-model $\mathbf{w}^n = \mathbf{w} \odot \mathbf{M}^n$, where $\odot$ denotes the element-wise multiplication and $\mathbf{M}^n$ is the mask. $\mathbf{M}^n_{l,i,j} = 0$ if the neuron $h_{l,i}$ of the parameter $w_{l,i,j}$ is pruned, and $\mathbf{M}^n_{l,i,j} = 1$ otherwise. The sampling probability of each neuron is determined based on its activation. We apply a softmax function over the activation $h_i$ of each neuron $i$, obtaining its sampling probability $p(i) = \frac{e^{h_i/T}}{\sum_{j=1}^m e^{h_j/T}}$, where $T$ is the temperature. Obviously, one neuron is more likely to be sampled once its activation is larger. In particular, the neurons are selected in a uniform manner as the temperature $T \to \infty$, while the neurons are selected in a TopK manner as the temperature $T \to 0$, i.e., selecting neurons with the highest activation values $\|h_{l,i}\|$.

The client locally updates the sub-model $\mathbf{w}^n = \mathbf{w}^n - \eta \nabla_{\mathbf{w}^n} f_n(\mathbf{w}^n)$, where $f_n(\mathbf{w}^n)$ denotes the loss over a mini-batch of data and $\eta$ is the learning rate. Then, the server receives the sub-models from all clients and aggregates them to update the global model: $\mathbf{w} = \mathbf{w} - \eta \sum_{n \in N} \mathbf{p}^n \odot \sum_{e=1}^E \nabla_{\mathbf{w}^n_e} f_n(\mathbf{w}^n_e)$, where $N$ denotes the set of selected clients and $\mathbf{p}^n$ endows a weight for each element of the sub-model parameters. We set $\mathbf{p}^n_{l,i,j} = \frac{1}{|N_{l,i,j}|}$ with $N_{l,i,j}$ representing the clients set that select the parameter $w_{l,i,j}$. In fact, the extraction process can also be conducted on the server by using a data-free manner like [47]. We leave the discussion in Appendix A.3.

## 6 THEORETICAL ANALYSIS

In this section, we formally analyze the performance of our proposed method compared to existing methods. We first show that our method achieves a higher probability score than existing methods over the two-layer neural networks with ReLU activation function. Then, we establish the convergence theory of our method over general non-convex loss functions.

### 6.1 Improved Probability Score

Following Theorem 1 and Proposition 2, we further compare the impact of neuron competition over the activation, i.e., reduced activation value by allocating positive neurons of some specific client to another heterogeneous client), and the probability score.

---

**Algorithm 1** FedDSE Algorithm
___
**Input:** Global model $\mathbf{w}$, and learning rate $\eta$, total communication rounds $T$.
**Output:** Trained global model $\mathbf{w}$.
1: Initialize the model parameters $\mathbf{w}_1$;
2: **procedure** SERVER-SIDE OPTIMIZATION
3:     **for** each communication round $t \in \{1, 2, ..., T\}$ **do**
4:         Randomly select a subset of clients $N_t$;
5:         Distribute $\mathbf{w}_t$ to each selected client;
6:         **for** each selected client $n$ **in parallel do**
7:             $\mathbf{w}^n_{t+1} \leftarrow ClientLocalUpdate(n, \mathbf{w}^n_t)$;
8:         Update the global model $\mathbf{w}_t = \mathbf{w}_t - \eta \sum_{n \in N_t} \mathbf{p}^n_t \odot \sum_{e=1}^E \nabla_{\mathbf{w}^n_{t,e}} f_n(\mathbf{w}^n_{t,e})$;
9: **procedure** CLIENTLOCALUPDATE$(n, \mathbf{w}^n_t)$
10:     Receive $\mathbf{w}_t$ from the server;
11:     Sample $r$ neurons layer-by-layer in activation-based probability to obtain the sub-model $\mathbf{w}^n_{t,1} = \mathbf{w}_t \odot \mathbf{M}^n_t$;
12:     **for** each local iterations $e$ from 1 to $E$ **do**
13:         Update sub-model parameters on private data $\mathbf{w}^n_{t,e+1} = \mathbf{w}^n_{t,e} - \eta \nabla_{\mathbf{w}^n_{t,e}} f_n(\mathbf{w}^n_{t,e})$;
        **return** Local update of the sub-model $\sum_{e=1}^E \nabla_{\mathbf{w}^n_{t,e}} f_n(\mathbf{w}^n_{t,e})$;
___

PROPOSITION 3. *When training sub-models on clients with heterogeneous distributions relative to a specific client $n$, the reduction in neuron activation $\Delta h(\mathbb{D}_n)$ for a two-layer neural network over the data $\mathbb{D}_n$ of the specific client, achieved through either random or sequential neuron selection strategies, is greater compared to that of our distribution-aware selection method $\Delta h'(\mathbb{D}_n)$ under the worst-case, i.e., $\Delta h(\mathbb{D}_n) \geq \Delta h'(\mathbb{D}_n)$.*

The proof can be found in B. The key is that existing strategies cannot avoid allocating the top neurons of some specific client to the other clients with heterogeneous distribution to the client, leading to a great activation reduction to these top neurons. Then, we have the following theory to show that the probability score will also be reduced due to the reduced activation activation.

THEOREM 4. *Given a two-layer converged neural network including $m$ neurons with the ReLU activation function. The obtained probability score $p_s(\mathbb{D}_n)$ over the dataset $\mathbb{D}_n$ of some specific client $n$ for a given class $s$, after running on heterogeneous clients with sub-models extracted through either random or sequential neuron selection strategies, is smaller than the probability score of our distribution-aware selection method $p'_s(\mathbb{D}_n)$ under the worst-case, i.e., $p_s(\mathbb{D}_n) \leq p'_s(\mathbb{D}_n)$.*

The proof is in B. Theorem 4 indicates that our method can maintain the probability score of previous clients by avoiding allocating neurons to conflicted clients with heterogeneous distributions. Hence, our method can help the global model memorize the data of clients selected in old rounds and improve the training accuracy.

### 6.2 Convergence Analysis

To show the convergence, we make the following assumptions which are widely adopted in FL.

ASSUMPTION 1. *(L-smoothness). The objective function $F$ is continuously differentiable and the gradient function of $F$ is L-smooth*

with Lipschitz constant $L_s > 0$, i.e., for all $\mathbf{w}$, $\mathbf{w}'$,

$$\|\nabla F(\mathbf{w}) - \nabla F(\mathbf{w}')\|_2 \le L_s \|\mathbf{w} - \mathbf{w}'\|_2.$$

ASSUMPTION 2. *(Bounded variance). For all parameters $\mathbf{w}$, the variance of the stochastic gradient in each client is bounded:* $\mathbb{E}(\|\nabla_\mathbf{w} f_n(\mathbf{w}) - \nabla_\mathbf{w} F_n(\mathbf{w})\|^2) \le \sigma^2$.

ASSUMPTION 3. *(Bounded Hessian). There exists positive a constant $H$ such that for all $\mathbf{w}$ and $n$, the second partial derivatives of $f_n$ with respect to the activation $h_{n,l,i}$ for each layer $l$ and neuron $i$ satisfy:* $\|\nabla^2_{h_{n,l,i}} f_n(\mathbf{w})\|^2 \le H$.

ASSUMPTION 4. *(Bounded Gradient). For all parameters $\mathbf{w}$, the gradient with respect to the loss is bounded:* $\mathbb{E}(\|\nabla_\mathbf{w} f_n(\mathbf{w})\|^2) \le G^2$, *and the embedding gradient with respect to each $i$-th neuron in the $l$-th layer is also boundded* $\mathbb{E}(\|\nabla_{\mathbf{w}_{l,i}} h_{l,i}(\mathbf{w})\|^2) \le G_h^2$.

The first two assumptions are generally used in the standard analysis of Federated Learning [12, 40, 43]. Based on these assumptions, we derive the convergence properties of our algorithm on general neural networks with ReLU activation function. The third assumption is a strengthened version of Assumption 1, which is also leveraged by previous studies [8]. The assumption of the bounded gradient regarding the loss is also generally utilized [45]. Assumption 4 slightly strengthens traditional assumption by also assuming the bounded gradient regarding the activation.

To simplify analysis, we introduce an iteration index $k$ where $k = t * E + e$. We also introduce an auxiliary model $\hat{\mathbf{w}}_k^n$, which is the full model obtained by filling the sub-model $\mathbf{w}_k^n$ with global parameters in the latest global round. Notably, according to the updating formula, $\hat{\mathbf{w}}_k^n = \mathbf{w}_t$ when $k = t * E$. To measure the impact of extracting neurons. We define the error between the activation $\mathbf{h}_{m,k}^n$ computed from the sub-model $\mathbf{w}_k^n$ and $\mathbf{h}_k^n$ calculated from the filled auxiliary model $\hat{\mathbf{w}}_k^n$, as $\mathbf{e}_k^n = \mathbf{h}_{m,k}^n - \mathbf{h}_k^n$. Based on these definitions, we then have the following lemma.

LEMMA 1. *The error of the gradient calculated by the sub-model is bounded by*

$$\mathbb{E}\|\nabla_{\hat{\mathbf{w}}_k^n} f_n(\hat{\mathbf{w}}_k^n) - \nabla_{\mathbf{w}_k^n} f_n(\mathbf{w}_k^n)\|_2^2 \le G_h^2 H^2 \sum_{l=1}^{L} \mathbb{E} \sum_{i \in S_{l,k}} \|\mathbf{e}_{l,i,k}\|_2^2$$

$$+ \sum_{l=1}^{L} \mathbb{E} \sum_{i \in S_{l,k}^c} \|\nabla_{\hat{\mathbf{w}}_{l,i,k}^n} f_n(\hat{\mathbf{w}}_k^n)\|_2^2, \qquad (2)$$

*where $S_{l,\tau}$ is the set of selected neurons in the $l$-th layer and $S_{l,\tau}^c$ denotes the set of un-selected neurons. $\hat{\mathbf{w}}_{l,i,k-1}^n$ represents the parameters connected to the neuron $i$.*

The proofs are deferred to Appendix C.1. Lemma 1 indicates that the error of the gradient calculated by the sub-model is related to the activation difference and the gradient unselected by the sub-model. Based on this lemma, we can derive the following theorem for the convergence of the algorithm.

THEOREM 5. *Considering $F_*$ be the global minima of the loss function, $\gamma$ and $\alpha$ are constants with $\gamma > 0$, $0 \le \alpha < 1$, and the learning rate $0 < \eta \le \frac{1}{L_s}$, then for all neural networks with ReLU activation function, the expected average of the squared gradient*

norms of $F$ obtained by Algorithm 1 satisfies the following bound for all $K \in \mathbb{N}$:

$$\sum_{k=1}^{K} \mathbb{E}\|\frac{1}{N}\sum_{n=1}^{N} \nabla_{\hat{\mathbf{w}}_k^n} F_n(\hat{\mathbf{w}}_k^n)\|_2^2 \le \frac{2(F(\mathbf{w}_1) - F_*)}{\eta}$$

$$+ 2L_s^2 \eta^2 \alpha G^2 (1+\gamma) \frac{(1+\gamma)^{K-1} - 1}{\gamma^2} + \frac{KL_s \eta \sigma_2^2}{N}$$

$$+ 4L_s^2 \eta^2 (1 + \frac{1}{\gamma}) G_h^2 H^2 \sum_{k=1}^{K} \sum_{\tau=1}^{k-1} (1+\gamma)^{k-1-\tau} \sum_{l=1}^{L} \mathbb{E} \sum_{i \in S_{l,\tau}} \|\mathbf{e}_{l,i,\tau}\|_2^2$$

$$+ 16KL_s^2 \eta^4 E^2 G^2 (1 + \frac{1}{\gamma}), \qquad (3)$$

*where $S_{l,\tau}$ is the set of selected neurons in the $l$-th layer.*

Detailed derivations are deferred to Appendix C.2. Theorem 5 shows that the convergence performance of FL with sub-model extraction heavily relies on the activation error $\mathbf{e}_k^n$. Rather than selecting neurons based on their location according to conventional methods, our approach extracts neurons based on the magnitude of their activation. Hence, our method maximizes the potential to reduce the activation error. Since the global model $\mathbf{w}_t$ periodically equals $\hat{\mathbf{w}}_k^n$, Theorem 5 also indicates the convergence of the global model, i.e., $\sum_{t=1}^{T} \mathbb{E}\|\nabla_{\mathbf{w}_t} F(\mathbf{w}_t)\|_2^2 \le \sum_{k=1}^{K} \mathbb{E}\|\frac{1}{N}\sum_{n=1}^{N} \nabla_{\hat{\mathbf{w}}_k^n} F_n(\hat{\mathbf{w}}_k^n)\|_2^2$. Now, we consider the feature distance $\|\mathbf{e}\|_2^2$ is bounded by a constant $\epsilon > 0$ which is determined by the ratio, i.e., $\|\mathbf{e}\|_2^2 \le \epsilon^2$. Obviously, $\epsilon \to 0$ as $r \to 1$. We show that the final convergence error is strongly related to the extraction ratio $r$.

THEOREM 6. *Considering $F_*$ be the global minima of the loss function and the learning rate $0 < \eta \le \frac{1}{4L_s}$, then for all neural networks with ReLU activation function, the expected average of the squared gradient norms of $F$ obtained by Algorithm 1 satisfies the following bound for all $t \in \mathbb{N}$:*

$$\frac{1}{T} \sum_{t=1}^{T} \|\nabla_{\mathbf{w}_t} F(\mathbf{w}_t)\|_2^2 \le \frac{4(F(\mathbf{w}_1) - F_*)}{\sqrt{T}}$$

$$+ 4E\left(\frac{L_s}{\sqrt{T}} + \frac{1}{2}\right)(G_h^2 H^2 r M \epsilon^2 + \alpha G^2) + \frac{8EL_s E^2 G^2}{T}, \qquad (4)$$

*where $\alpha$ is a constant relying on the extraction ratio of the sub-model with $0 \le \alpha < 1$.*

Proof can be found in Appendix C.3. Since $\epsilon \to 0$ and $\alpha \to 0$ as $r \to 1$, Theorem 6 indicates that the error asymptotically converges to 0 with respect to the iteration $t$ and $r$.

# 7 EXPERIMENTS

**Datasets and models.** We evaluate the performance of the proposed FedDSE over two models and three mainstream datasets. In specific, two distinct models including a CNN for EMNIST [24], a pre-activated ResNet [38] for CIFAR-10 and CIFAR-100 [23] are adopted for performance evaluation. The Static Batch Normalization method is applied instead of Batch Normalization, and a scalar module follows each convolution layer [10] is introduced. We use four convolution layers to compose the CNN model, whose channels are $\{64, 128, 256, 512\}$, respectively.

**Data heterogeneity.** For EMNIST, CIFAR-10 and CIFAR-100, we follow the non-IID split method in HeteroFL [10]. In the following

Table 1: The comparison of test accuracy of different methods. Each experiment is conducted three times with random seeds.

| Method | High Data Heterogeneity(%) | | | Low Data Heterogeneity(%) | | |
|---|---|---|---|---|---|---|
| | EMNIST | CIFAR-10 | CIFAR-100 | EMNIST | CIFAR-10 | CIFAR-100 |
| HeteroFL | 93.21±1.23 | 38.13±1.91 | 8.00±2.45 | 97.61±1.02 | 47.01±1.34 | 11.16±2.02 |
| Federated Dropout | 87.96±2.11 | 50.16±2.63 | 10.47±2.87 | 97.63±1.92 | 58.16±2.26 | 16.21±2.10 |
| FedRolex | 91.41±1.15 | 55.61±1.62 | 14.02±1.90 | 98.61±0.98 | 68.04±1.34 | 20.81±1.18 |
| FedDSE | **95.34**±1.24 | **58.19**±1.57 | **16.61**±1.87 | **98.65**±1.01 | **70.82**±1.16 | **22.93**±1.31 |

of this paper, $L$ indicates the number of classes each client has. According to the size of $L$, we define High Data Heterogeneity and Low Data Heterogeneity. For EMNIST and CIFAR-10, $L = 2$ indicates High Data Heterogeneity, and $L = 4$ means Low Data Heterogeneity For CIFAR-100, we adopt $L = 5$ for High Data Heterogeneity and $L = 10$ for Low Data Heterogeneity.

**Model heterogeneity.** We define five different client model capacities $\beta$ ={1 (0, 0.01, 0.99), 1/2 (0.01, 0.98, 0.01), 1/4 (0.01, 0.98, 0.01), 1/8 (0.01, 0.98, 0.01), 1/16 (0, 1, 0)}. As most clients' capacities do not reach the capacity of the server and include several intermediate values, we define a ratio $\alpha = 1/16$ to better simulate the real client distribution. Each client's model capacity fluctuates around $\alpha$ of the original capacity. Using 1/2 as an example, 1/2 represents client model capacity. (0.01, 0.98, 0.01) i.e., the probability distribution of {1/2 + 1/16, 1/2, 1/2 − 1/16}. The global model channels are allocated according to the number of channels in each layer of the client model.

**Baselines.** We compare three Partial Training (PT)-based FL methods. Specifically, HeteroFL [10] is a static distribute neuron method. FedRolex [39] and Federated Dropout[7] are dynamic distribute neuron methods. To guarantee the fairness of comparison, we use the same learning rate, local epochs, as well as communication rounds. In this paper, we mainly focus on the performance of FedDSE rather than model optimization using the existing multi-step learning rate decay schedule that may lead to an efficiency decrease. More details about each method and dataset can be found in the Appendix D (including the setting of Table 2-5).

**Configurations and platform.** For EMNIST, CIFAR-10 and CIFAR-100, we apply bounding box crop [34] to augment the images. In each communication round, 10% of the 100 clients are selected for training, with frc = 10%. At the beginning of each communication round, the selected clients' capacities are dynamically chosen from a uniform distribution. Experiments are conducted atop PyTorch framework. The specifics of hyperparameters are shown in the Appendix. Experiments are carried out on computing machines with Nvidia RTX 3090, K80 and 1080Ti GPUs.

**Evaluation metric.** For image classification tasks, global accuracy is adopted as the evaluation metric, which is defined as the server model's accuracy over the entire test set. Besides, we also compare the cost of memory, communication, and computation of FedAvg and FedDSE in Table 7 of the appendix.

### 7.1 Performance Comparison with Baselines

Table 1 compares our FedDSE with four PT-base methods. The temperature of FedDSE is set to be 0. For a fair comparison, the client distribution is done in the aforementioned way. We observe that FedDSE achieves the best performance over the other three methods

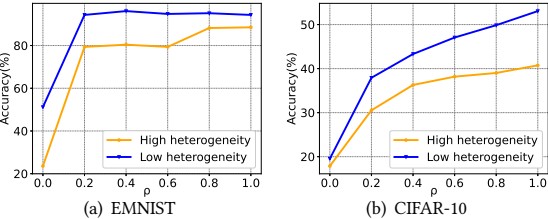

(a) EMNIST                    (b) CIFAR-10

Figure 4: Impact of client model heterogeneity distribution in EMNIST and CIFAR-10

under both high data heterogeneity and low data heterogeneity conditions. In addition, the results have proved that under high data heterogeneity, FedDSE significantly outperforms FedRolex on EMNIST and CIFAR10. This indicates that when the number of classes is relatively small, our method can accurately capture and activate the relevant neurons for training, hence achieving better results on EMNIST and CIFAR10 with 10 classes and $L = 2$. While for CIFAR100 with 100 classes and $L = 5$ where the sizes of the client dataset remain the same, it becomes difficult to select the active neurons, and the improvement is a mere 0.9%. Under low data heterogeneity where the client datasets are evenly distributed, the model converges faster and leads to prominent training overhead reduction. On the simple EMNIST dataset, FedDSE achieves similar accuracy as FedRolex. For complex datasets like CIFAR10 and CIFAR100, under more evenly distributed data, FedDSE outperforms other methods significantly by selecting and activating relevant neurons. HeteroFL can hardly cope with the situation when most client capacities are not up to the server capacity. The reason is that the neurons in the later part of the same layer will be trained with few times, and these neurons cause an accuracy drop in the global model. This phenomenon is not very obvious over EMNIST due to the simplicity of the dataset, as training a limited number of neurons can achieve decent results. The Federated Dropout method performs moderately. It randomly drops neurons causing high variance and instability. The performance of FedRolex is second only to FedDSE. We have thoroughly analyzed the reasons in the theoretical part, so we omit it here.

### 7.2 Impact of Client Model Heterogeneity

In the above experiments, the distribution of client capacities is set uniformly. Now we conduct the test by varying the value of $\rho$ to introduce different distributions. We choose two client model capacities $\beta$=1/2,1/16. $\rho$ is defined as the proportion of 1/2 clients. For example, $\rho = 0.2$ means that client capacity of 1/2 accounts for 0.2 and 1/16 accounts for 0.8.

Figure 5 shows that the accuracy increases as $\rho$ increases on the whole. For EMNIST in Figure 4(a), under high data heterogeneity,

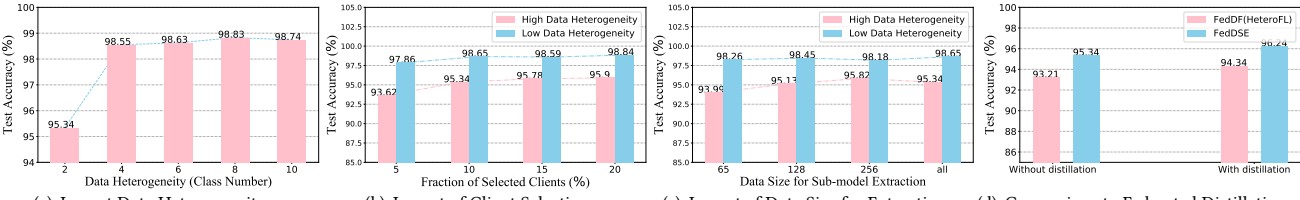

(a) Impact Data Heterogeneity    (b) Impact of Client Selection    (c) Impact of Data Size for Extraction    (d) Comparison to Federated Distillation

**Figure 5: Ablation Study**

the peak is reached at $\rho$=1. This indicates that the model convergence requires a combination of a large number of models. Thus the accuracy increases linearly with $\rho$. Under low data heterogeneity, the peak appears at $\rho$ = 0.4, proving that a large global model is not a prerequisite for fast convergence. Therefore, when $\rho$ exceeds 0.4, the model accuracy fluctuates up and down as $\rho$ increases. For the complex CIFAR-10 dataset in Figure 4(b), the accuracy continues to increase with the increase of $\rho$. This indicates that FedDSE is suitable for appropriately increasing the model parameters to improve the effect when dealing with complex problems.

### 7.3 Impact of Statistical Heterogeneity

In the above experiments, we define high and low data heterogeneity. In EMNIST, they are set as $L = \{2, 4\}$, respectively. Here, we set $L = \{2, 4, 6, 8, 10\}$. In doing so, the testing results can reflect the influence of the degree of data heterogeneity on global accuracy. Figure 5(a) shows that the accuracy improves significantly when $L = 2$ and $L = 4$, while the impact of data heterogeneity becomes mild from $L = 4$ to $L = 10$. In the scenario of 10 classes, it is common for users to encounter up to 4 classes at most.

### 7.4 Impact of Client Selection

Rather than simply setting frc as 10%, we vary the number of selecting clients from 5% to 20% with a step length of 5%. Figure 5(b) shows that under high data heterogeneity, frc improves the accuracy significantly when it increases from 5% to 10%. However, from 10% to 20%, the effect of frc becomes mild. Through Figure 5(b) we can find that a decent balance between model accuracy and convergence overhead can be reached when frc = 10%.

### 7.5 Impact of Data Size for Extraction

In the above experiments, the entire client dataset is adopted as the inference data. Here, we vary the inference batch size as $\{64, 128, 256, all\}$ to explore the impact of the inference data scale. In specific, 'all' refers to the size of the local dataset, which is 500 in EMNIST. Figure 5(c) shows that when the inference batch size reaches 128, the activated neurons selected can basically meet the requirements during inference. Figure 5(c) also indicates that simply increasing the inference batch size beyond 128 brings negligible accuracy gain. In other words, adopting an appropriate batch size leads to faster model convergence and fewer selected clients.

### 7.6 Comparison with Federated Distillation

FL with knowledge distillation accommodates heterogeneous model structures among clients and thus also allows training heterogeneous sub-models over different clients [31, 42]. In fact, *our method is orthogonal to these methods*. We can utilize FedDSE to extract sub-models and then adopt federated distillation to aggregate all sub-models. To show this, we also compare our method with FedDF [31]

on EMNIST, as shown in Figure 5(d). It can be observed that combining with federated distillation can further improve the performance of FedDSE. Besides, our method combined with federated distillation outperforms the baseline.

### 7.7 Impact of Temperature

In practice, we can also choose the temperature adaptively to achieve both benefits of activation-based selection and evenly-trained selection. To show this, we also conduct some experiments to compare FedDSE with hard-TopK and with soft-TopK, as shown in Table 2 on EMNIST. Homo. (1/4) denotes that all clients are homogeneous and can only train 1/4 of the full model, and Heterogeneous capability adopts the same setting as Table 1. It can be observed from the table that $T = 0$ and $T = 1$ perform better separately in different scenarios. Generally, higher temperature is more applicable to the settings where the capability of clients are homogeneous and vice versa. It is also worthwhile to note that our method always outperforms SOTA baseline, i.e., FedRolex.

**Table 2: Impact of different Temperature.**

| Capacity | Method | Data heterogeneity | | |
| | | High | Low | Homogeneity |
|---|---|---|---|---|
| Homo. (1/4) | FedRolex | 93.35 | 97.29 | 97.04 |
| | FedDSE (T=0) | 81.25 | 89.74 | 88.05 |
| | FedDSE (T=1) | **96.59** | **98.21** | **97.83** |
| Homo. (1/2) | FedRolex | 97.76 | 98.52 | 98.74 |
| | FedDSE (T=0) | 91.51 | 96.53 | 95.24 |
| | FedDSE (T=1) | **98.45** | **99.16** | **99.09** |
| Heterogeneous | FedRolex | 91.41 | 98.61 | 98.67 |
| | FedDSE (T=0) | **95.34** | **98.65** | **98.69** |
| | FedDSE (T=1) | 94.60 | 97.86 | 98.15 |

## 8 CONCLUSION

This paper focuses on sub-model extraction in federated learning. We have observed that clients tend to activate distinct neurons of the model due to statistical heterogeneity. This may lead to a competition problem for neurons in the sub-model when extracted inappropriately. To address this challenge, we propose a new sub-model extraction method for FL called FedDSE that exploits the activation distribution properties of neural networks and edge devices. Our method selects neurons with the largest activation value, adaptively designating them to different clients. We prove the convergence of our method theoretically and demonstrate its effectiveness through experimental results which outperform state-of-the-art techniques.

However, our method requires downloading the entire model which may increase communication costs. Furthermore, the local sub-model extraction process incurs extra computational costs despite being only an inference process. In addition to memory efficiency, we aim to further improve the efficiency in terms of communication and computation in future work.

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

## A MORE DETAILS ABOUT THE DESIGN OF MOTIVATION AND METHOD

### A.1 Investigation on Discrepancy of Activation Distribution

To investigate the principle behind the discrepancy of activation distribution among clients, we conduct the following experiment on a three-layer MLP with the MNIST dataset. The results are shown in Figure 6. There are a total of 5 clients and each client is allocated with 2 classes. The activation values outputted by the second layer over the datasets of all clients are illustrated together to demonstrate the difference among clients. Each client is denoted by a distinct color. It can be observed that the differences in data distribution can lead to disparities in the distribution of activation values.

The reason for activation reflecting data distribution is that activations between classes can be distinctly differentiated. The inference of neural networks is to progressively increase the linear separability of activation between classes from shallow to deep layers such that the last fully-connected classification layer can distinguish them. Hence, the activation can reflect its corresponding class and different classes also correspond to different activation. In FL, clients are usually equipped with various distributions of classes, and thus the activation distributions of different clients also differ from each other.

### A.2 More details about motivation

In the main text, we have shown that different clients (client 1 and client 2) with different data distributions tend to activate different neurons. For the comprehensiveness of this conclusion, we present the comparison results of all clients, as shown in Figure 7 and 8. Obviously, all client pairs will activate different neurons.

### A.3 More details about the design of method

The goal of FedDSE downloading the entire global model is to utilize the local dataset to identify neurons with large activation. In fact, many recent data-free methods have been proposed, which makes it unnecessary to rely on the real local dataset. Like [47], the server can train a generator based on the local model uploaded by each client. Then, the server utilizes the generator to produce pseudo-data samples which follow the same distribution as the local dataset. Based on these pseudo samples, the server can extract neurons from the global like FedDSE. One concern may be that the samples produced by the generator may cause privacy leakage which recovers the original samples. In fact, the recovery level heavily relies on the training strategies of the generator. The server can simply adopt the naive training method and learn the distribution instead of the original data samples. Besides, the generator can also be trained to generate intermediate feature maps instead of the original data samples to protect privacy.

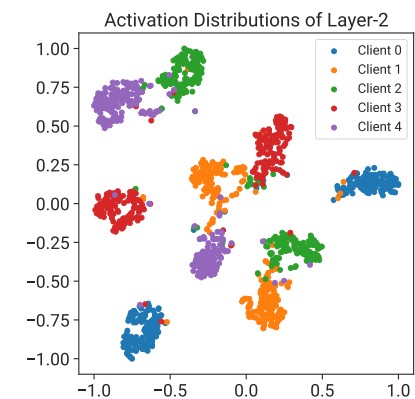

**Figure 6: Illustration of activation distribution of different clients.**

## B PROOFS OF THEORIES OVER THE TWO-LAYER NEURAL NETWORKS

To verify our motivation, we seek to first show that the classification accuracy is strongly related to the activation magnitude of neurons. Then, we show that the activation magnitude of some neurons of one client can be reduced by another client with different distributions. We consider a two-layer neural network with the popular ReLU as the activation function and there are $m$ neurons in the hidden layer. The neural network is trained with a basic cross-entropy loss function. Without losing generality, we mainly consider the binary classification task.

We denote the parameters of the second layer as $\mathbf{w}_2$ and the parameters of class $c$ are $\mathbf{w}_{2,c}$. Similarly, we denote $\mathbf{w}_{1,i}$ as the first-layer parameters corresponding to the $i$-th hidden neuron, and $w_{1,i,j}$ as a first-layer parameter connected between the input neuron $j$ and hidden neuron $i$. Besides, the activation of the $i$-th hidden neuron is denoted as $h_i^k = \sigma(\mathbf{w}_{1,i}\mathbf{x}^k)$ with the input sample as $\mathbf{x}^k$ with extend dimension 1 to incorporate the bias, and $\mathbf{h}^k = [h_i^1, h_i^2, \ldots, h_i^m]$ as the activation vector outputted by the hidden layer.

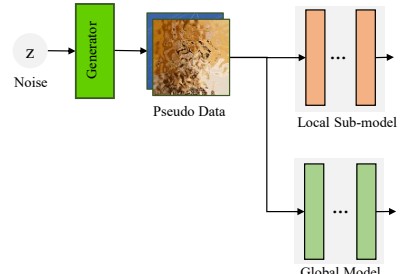

**Figure 9: Server extracts sub-models based on the pseudo data.**

To optimize the parameters, the neural network will first compute the probability for each class $c$ and sample $\mathbf{x}^k$:

$$p_c^k = \frac{e^{\mathbf{w}_{2,c}\mathbf{h}^k}}{\sum_{s=1}^{C} e^{\mathbf{w}_{2,s}\mathbf{h}^k}}, \tag{5}$$

and the cross-entropy loss is:

$$L_k = -\sum_{c=1}^{C} I(y^k = c)\log p_c^k, \tag{6}$$

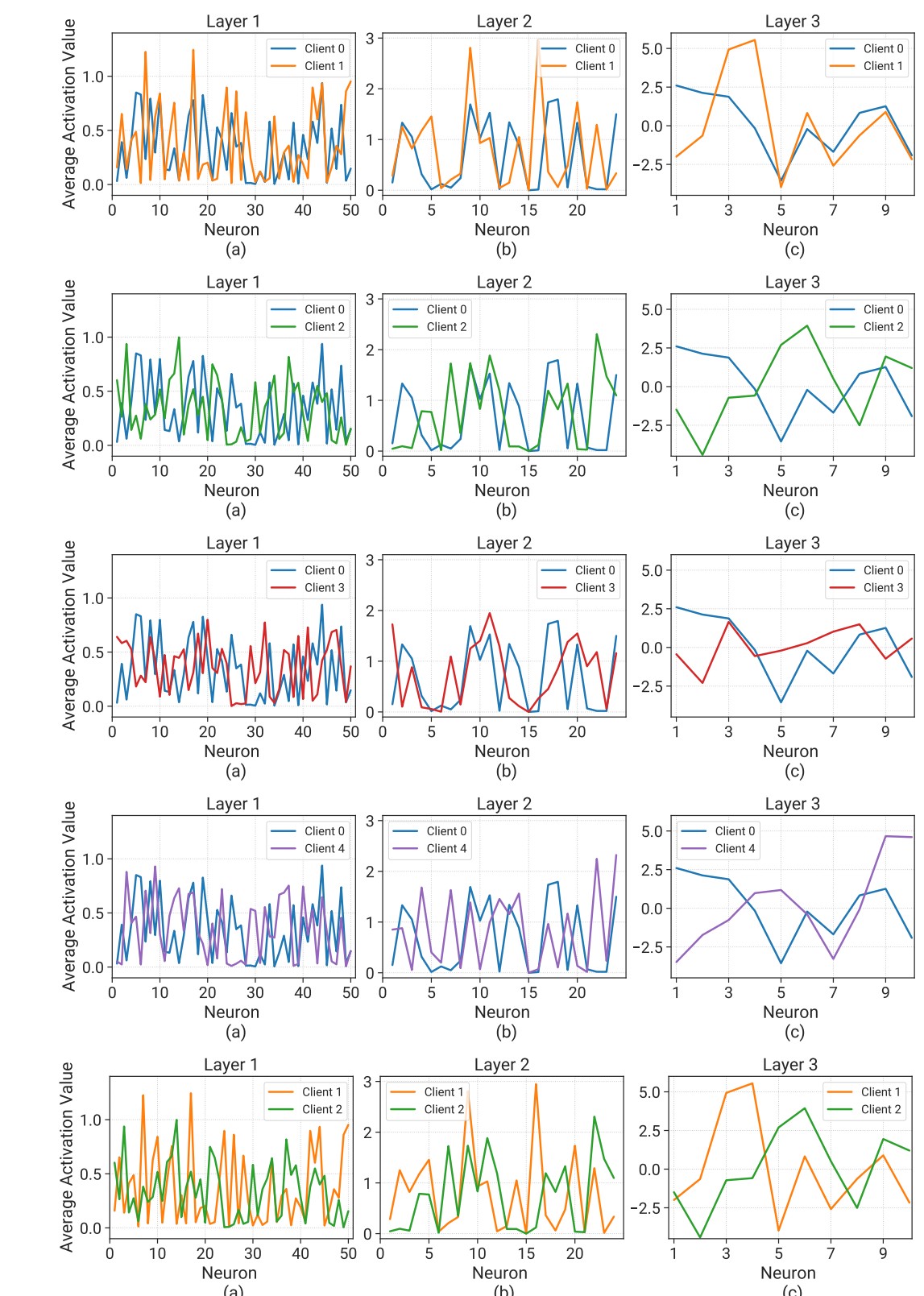

**Figure 7: Comparison of activation distributions of a 3-layer MLP on MNIST. (a-c) Activations of two clients on layer-1, 2 and 3.**

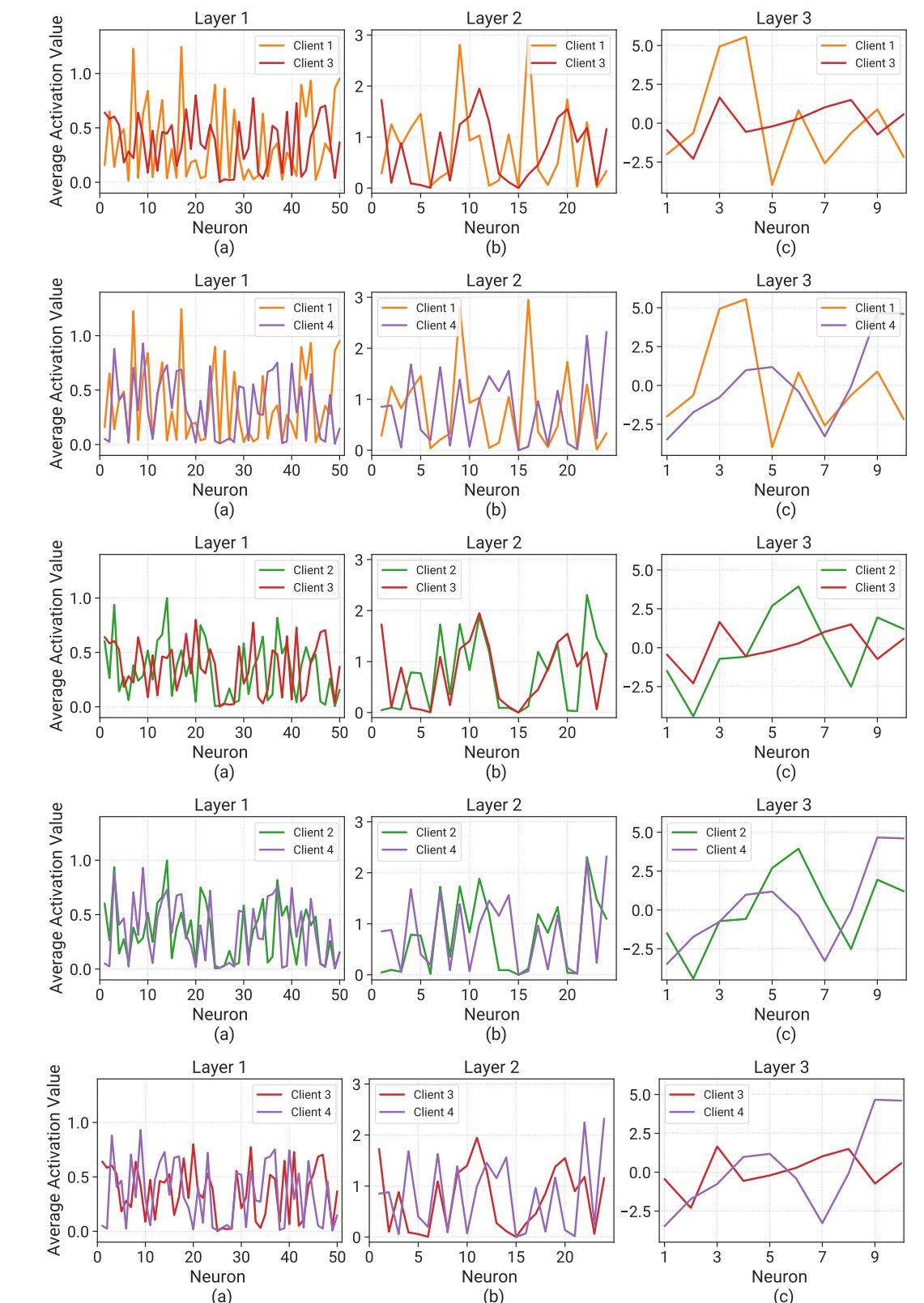

**Figure 8: Comparison of activation distributions of a 3-layer MLP on MNIST. (a-c) Activations of two clients on layer-1, 2 and 3.**

where $I(\cdot)$ denotes indication function. The gradient of $\mathbf{w}_c^k$ is:

$$g(\mathbf{w}_{2,c}^k) = -(I(y^k = c) - p_c^k)\mathbf{h}^k. \tag{7}$$

According to the process of backward propagation, the gradient $g(\mathbf{w}_{1,i}^k)$ of the parameter corresponding to the $i$-th hidden neuron is:

$$g(\mathbf{w}_{1,i}^k) = -\left[\sum_{c=1}^{C}(I(y_k = c) - p_c^k)w_{2,c,i}\right]\mathbf{x}^k$$

$$= -(1 - p_c^k)w_{2,c,i}\mathbf{x}^k + \sum_{s \neq c, s=1}^{C} p_s^k w_{2,s,i}\mathbf{x}^k \tag{8}$$

Next, we show that the activation magnitude of some neurons of one client can be reduced by another client with different distributions.

THEOREM 1. *Consider a two-layer neural network employing the ReLU activation function and being trained with a cross-entropy loss. Let $\mathbb{D}_{n_1}$ comprise samples belonging to class $s$, and $\mathbb{D}_{n_2}$ consist of samples from class $c$, representing the datasets of clients $n_1$ and $n_2$ respectively. Let $h_i(\mathbb{D}_{n_1}) = \sum_{j=1}^{D} ReLU(\mathbf{w}_i^T\mathbf{x}^j)$ represent the sum of activations of the $i$-th selected hidden neuron across dataset $\mathbb{D}_{n_1}$, with $D$ denoting the dataset size. Subsequently, training the sub-model $\hat{\mathbf{w}}$ on dataset $\mathbb{D}_{n_2}$ and denoting $p_s^k$ as the probability score of sample $\mathbf{x}^k \in \mathbb{D}_{n_2}$ over the trained sub-model, with a learning rate $\eta > 0$, yields the following observations:*

• *When the dataset $\mathbb{D}_{n_1}$ of client $n_1$ is homogeneous to the local training dataset $\mathbb{D}_{n_2}$ of client $n_2$, i.e., $\sum_{\mathbf{x}^k \in \mathbb{D}_{n_2}} p_s^k(\mathbf{x}^k)^T\mathbf{x}^j \geq 0$ for each sample $\mathbf{x}^j \in \mathbb{D}_{n_1}$, the activation sum $h_i(\mathbb{D}_{n_1})$ increases, where the augmentation can be as high as $\eta \sum_{\mathbf{x}^j \in \mathbb{D}_{n_1}} \sum_{\mathbf{x}^k \in \mathbb{D}_{n_2}} p_s^k(w_{2,c,i} - w_{2,s,i})(\mathbf{x}^k)^T\mathbf{x}^j$.*

• *Conversely, when the dataset $\mathbb{D}_{n_2}$ of client $n_1$ is heterogeneous to the local training dataset $\mathbb{D}_{n_2}$ of client $n_2$, i.e., $\sum_{\mathbf{x}^k \in \mathbb{D}_{n_2}} p_s^k(\mathbf{x}^k)^T\mathbf{x}^j \leq 0$ for each sample $\mathbf{x}^j \in \mathbb{D}_{n_1}$, the activation sum $h_i(\mathbb{D}_{n_1})$ decreases, where the reduction is $Min(-\eta \sum_{\mathbf{x}^j \in \mathbb{D}_{n_1}} \sum_{\mathbf{x}^k \in \mathbb{D}_{n_2}} p_s^k(w_{2,c,i} - w_{2,s,i})(\mathbf{x}^k)^T\mathbf{x}^j, h_i(\mathbb{D}_{n_1}))$.*

PROOF. To investigate the change of activation values over the previous client $n_1$ and current client $n_2$, we start with the optimization of the last-layer classifier parameters. Specifically, we consider the current client contains the samples of class $c$ whereas the previous client only contains the samples of class $s$. After neuron selection, we denote $N_i$ the set of selected neurons for each client $i$ and denote $\hat{\mathbf{h}}$ as the activation vector of the hidden layer in sub-model $\hat{\mathbf{w}}$. The gradient of parameters corresponding to the class $c$ and class $s$ respectively for each sample $\mathbf{x}^k$ with the label $y^k = c$ is:

$$\frac{\nabla L_k}{\nabla \hat{\mathbf{w}}_{2,c}^k} = -(1 - p_c^k)\hat{\mathbf{h}}^k = -\left(1 - \frac{e^{\hat{\mathbf{w}}_{2,c}\hat{\mathbf{h}}^k}}{\sum_{i=1}^{C} e^{\hat{\mathbf{w}}_{2,i}\hat{\mathbf{h}}^k}}\right)\hat{\mathbf{h}}^k,$$

$$\frac{\nabla L_k}{\nabla \hat{\mathbf{w}}_{2,s}^k} = p_s^k\hat{\mathbf{h}}^k = \frac{e^{\hat{\mathbf{w}}_{2,s}\hat{\mathbf{h}}^k}}{\sum_{i=1}^{C} e^{\hat{\mathbf{w}}_{2,i}\hat{\mathbf{h}}^k}}\hat{\mathbf{h}}^k. \tag{9}$$

The updating formula of the two parameters is:

$$\hat{\mathbf{w}}_{2,c}^k = \hat{\mathbf{w}}_{2,c}^k + \eta(1 - p_c^k)\hat{\mathbf{h}}^k, \quad \hat{\mathbf{w}}_{2,s}^k = \hat{\mathbf{w}}_{2,s}^k - \eta p_s^k\hat{\mathbf{h}}^k. \tag{10}$$

Since the activation value from the ReLU function is always positive, i.e., $\mathbf{h} \overset{\geq}{} 0$, we can intuitively find that the parameters $\hat{\mathbf{w}}_{2,c}^k$ corresponding to the local class $c$ always increase while the parameters $\hat{\mathbf{w}}_{2,s}^k$ corresponding to the class $s$ of previous client always decreases. Further, we can derive the final converged parameter by solving the following equation to find the saddle points:

$$\frac{\partial L_k}{\partial \hat{\mathbf{w}}_{2,c}^k} = 0, \quad \frac{\partial L_k}{\partial \hat{\mathbf{w}}_{2,s}^k} = 0, \tag{11}$$

where the solution is:

$$\hat{\mathbf{w}}_{2,c}^k \to \infty, \quad \hat{\mathbf{w}}_{2,c}^k \to -\infty. \tag{12}$$

Hence, we can immediately derive that the local training process over all samples of local data will update the classifier parameters as

$$\hat{\mathbf{w}}_{2,c} \to \infty, \quad \hat{\mathbf{w}}_{2,c} \to -\infty. \tag{13}$$

Now, we investigate the update of parameters in the first layer. For each selected neuron $i \in N_{n_2}$, the gradient of its correspond parameters $\mathbf{w}_{1,i}^k$ for each sample $\mathbf{x}^k$ with label $y^k = c$ is:

$$\frac{\partial L_k}{\partial \mathbf{w}_{1,i}^k} = -\left[\sum_{c=1}^{C}(I(y_k = c) - p_c^k)w_{2,s,i}\right]\mathbf{x}^k$$

$$= -(1 - p_c^k)w_{2,c,i}\mathbf{x}^k + \sum_{s \neq c, s=1}^{C} p_s^k w_{2,s,i}\mathbf{x}^k$$

$$= p_s^k(-w_{2,c,i} + w_{2,s,i})\mathbf{x}^k \tag{14}$$

where $I(\cdot)$ is an indication function. By applying the local training process over the local dataset $\mathbb{D}_n$, we have the updated formula of the parameter $\mathbf{w}_{1,i}^k$ as:

$$\mathbf{w}_{1,i}' = \mathbf{w}_{1,i} + \eta \sum_{k=1}^{D} p_s^k(w_{2,c,i} - w_{2,s,i})\mathbf{x}^k, \tag{15}$$

where $D$ is the number of samples in each client. Based on equation (13), we can get that $w_{2,c,i} - w_{2,s,i} > 0$ when the number of local training epochs is sufficient. Now, we can obtain the activation average of this updated neuron $i$ over any dataset $\mathbb{D}$:

$$\begin{aligned}
h_i'(\mathbb{D}) &= \sum_{\mathbf{x}^j \in \mathbb{D}} \text{ReLU}(\mathbf{w}_{1,i}'\mathbf{x}^j) \\
&= \sum_{j=1}^{D} \text{ReLU}(\mathbf{w}_{1,i}^T\mathbf{x}^j + \eta \sum_{k=1}^{D} p_s^k(w_{2,c,i} - w_{2,s,i})(\mathbf{x}^k)^T\mathbf{x}^j),
\end{aligned} \tag{16}$$

When the dataset $\mathbb{D}$ is homogeneous to the local dataset $\mathbb{D}_n$, i.e., $\sum_{\mathbf{x}^k \in \mathbb{D}_n} p_s^k(\mathbf{x}^k)^T\mathbf{x}^j \geq 0$ for any $\mathbf{x}^j \in \mathbb{D}$, according to the convexity of monotonicity of the ReLU function, we have

$$\begin{aligned}
h_i(\mathbb{D}) &= \sum_{j=1}^{D} \text{ReLU}(\mathbf{w}_{1,i}^T\mathbf{x}^j) \leq h_i'(\mathbb{D}) \\
&= \sum_{j=1}^{D} \text{ReLU}(\mathbf{w}_{1,i}^T\mathbf{x}^j + \eta \sum_{k=1}^{D} p_s^k(w_{2,c,i} - w_{2,s,i})(\mathbf{x}^k)^T\mathbf{x}^j) \\
&\leq \sum_{j=1}^{D} \text{ReLU}(\mathbf{w}_{1,i}^T\mathbf{x}^j) + \sum_{j=1}^{D} \text{ReLU}(\eta \sum_{k=1}^{D} p_s^k(w_{2,c,i} - w_{2,s,i})(\mathbf{x}^k)^T\mathbf{x}^j) \\
&= h_i(\mathbb{D}) + \eta \sum_{j=1}^{D} \sum_{k=1}^{D} p_s^k(w_{2,c,i} - w_{2,s,i})(\mathbf{x}^k)^T\mathbf{x}^j,
\end{aligned} \tag{17}$$

where $h_i(\mathbb{D})$ represents the activation mean of the $i$-th neuron of the non-updated model $\hat{\mathbf{w}}$ over the dataset $\mathbb{D}$. Based on this equation (17), considering $\mathbb{D} = \mathbb{D}_n$, we can immediately derive that the local training process increases the neuron activation over the local dataset, i.e., $h_i(\mathbb{D}_n) \leq h_i'(\mathbb{D}_n)$. The increased overall activation is $\eta \sum_{j=1}^{D} \sum_{k=1}^{D} p_s^k(w_{2,c,i} - w_{2,s,i})(\mathbf{x}^k)^T\mathbf{x}^j$.

Similarly, when the dataset $\mathbb{D}$ is heterogeneous to the local dataset $\mathbb{D}_n$, i.e., $\sum_{\mathbf{x}^k \in \mathbb{D}_n} p_s^k(\mathbf{x}^k)^T\mathbf{x}^j \leq 0$ for any $\mathbf{x}^j \in \mathbb{D}$, according to the convexity of monotonicity of the ReLU function, we have

$$\begin{aligned}
h_i'(\mathbb{D}) &= \sum_{j=1}^{D} \text{ReLU}(\mathbf{w}_{1,i}^T\mathbf{x}^j + \eta \sum_{k=1}^{D} p_s^k(w_{2,c,i} - w_{2,s,i})(\mathbf{x}^k)^T\mathbf{x}^j) \\
&= \text{Max}(\sum_{j=1}^{D} \text{ReLU}(\mathbf{w}_{1,i}^T\mathbf{x}^j) - \sum_{j=1}^{D} \text{ReLU}(-\eta \sum_{k=1}^{D} p_s^k(w_{2,c,i} - w_{2,s,i})(\mathbf{x}^k)^T\mathbf{x}^j), 0) \\
&= \text{Max}(h_i(\mathbb{D}) + \eta \sum_{j=1}^{D} \sum_{k=1}^{D} p_s^k(w_{2,c,i} - w_{2,s,i})(\mathbf{x}^k)^T\mathbf{x}^j, 0) \\
&\leq \sum_{j=1}^{D} \text{ReLU}(\mathbf{w}_{1,i}^T\mathbf{x}^j) \\
&= h_i(\mathbb{D}).
\end{aligned} \tag{18}$$

Hence, the overall activation of each $i$-th selected neuron over the dataset $\mathbb{D}$ is reduced when the sub-model has been locally updated with the local dataset $\mathbb{D}_n$, i.e., $h_i(\mathbb{D}) \geq h_i'(\mathbb{D})$. The reduced overall activation is $\text{Min}(-\eta \sum_{j=1}^{D} \sum_{k=1}^{D} p_s^k(w_{2,c,i} - w_{2,s,i})(\mathbf{x}^k)^T\mathbf{x}^j, h_i(\mathbb{D}))$. The proof is done. □

*Proposition 2. Given a well-converged two-layer neural network with the ReLU activation function, high activation values have a large impact on the probability score than low activation values. Specifically, for any sample $\mathbf{x}$ with label $y = c$, the ratio of impact over probability score $p_c$ between a high activation $h_H$ and a low activation $h_L$ is approximately $e^{\alpha(h_H^2 - h_L^2)}$, where $\alpha > 0$ is a constant.*

*Proof*: Based on [1] (Theorem 1), all sample features of each class $c$ collapse to their mean $\mathbf{h}_c$ and the converged parameters $\mathbf{w}_{2,c}$ of the class $c$ have the same direction as the activation mean of its class, i.e., $\mathbf{w}_{2,c} = \alpha\mathbf{h}_c$, where $\alpha > 0$ is a constant. Hence, the activation of a given

sample $\mathbf{x}$ with label $y = c$ is approximately equivalent to the activation mean $\mathbf{h} \approx \mathbf{h}_c$. Accordingly, the impact of the high activation and low activation can be obtained separately based on equation (5) as

$$\text{Impact}(p_c, h_H) = \frac{e^{\mathbf{w}_{2,c,H} h_H}}{\sum_{s=1}^{C} e^{\mathbf{w}_{2,s} \mathbf{h}}} \approx \frac{e^{\alpha h_H^2}}{\sum_{s=1}^{C} e^{\mathbf{w}_{2,s} \mathbf{h}}}, \quad (19)$$

$$\text{Impact}(p_c, h_L) = \frac{e^{\mathbf{w}_{2,c,L} h_L}}{\sum_{s=1}^{C} e^{\mathbf{w}_{2,s} \mathbf{h}}} \approx \frac{e^{\alpha h_L^2}}{\sum_{s=1}^{C} e^{\mathbf{w}_{2,s} \mathbf{h}}}. \quad (20)$$

Computing the ratio between the two impacts derives the proposition.

[1] Fang, Cong, et al. "Exploring deep neural networks via layer-peeled model: Minority collapse in imbalanced training." Proceedings of the National Academy of Sciences 118.43 (2021): e2103091118.

*Proposition 3. When training sub-models on clients with heterogeneous distributions relative to a specific client, the reduction in neuron activation for a two-layer neural network, achieved through either random or sequential neuron selection strategies, is greater compared to that of our distribution-aware selection method under the worst-case.*

*proof* Based on Theorem 1, the activation magnitude of neurons over the dataset $\mathbb{D}_n$ of some specific client $n$ will be reduced to 0 under the worst-case when these neurons are allocated to another client with heterogeneous data distribution to this specific client. Since existing strategies cannot avoid allocating the top neurons of some specific client to other clients with heterogeneous distributions, they will reduce the activation of neurons with the highest magnitudes to 0. Denote the activation of $i$-th neuron over the specific client $n$ as $h_i^n(\mathbb{D}_n)$ and the selected $r$ neurons by other clients with the highest magnitudes are numbered from 1 to $r$. Then, the overall reduction in activation by existing strategies is

$$\Delta h = \sum_{i=1}^{r} h_i^n(\mathbb{D}_n). \quad (21)$$

Considering that our distribution-aware method avoids clients selecting the top neurons in the client $n$ when they have heterogeneous distributions, we denote the selected neurons to be $o^1, \ldots, o^r$. Hence, the overall reduction in activation by our method is

$$\Delta h' = \sum_{i=1}^{r} h_{o^i}^n(\mathbb{D}_n) \quad (22)$$

Considering that the neurons numbered 1 to $r$ have the largest activation, i.e.,

$$h_i^n(\mathbb{D}_n) \le h_j^n(\mathbb{D}_n), \text{for any} 1 \le i \le r, r+1 \le j \le m, \quad (23)$$

we have

$$\Delta h = \sum_{i=1}^{r} h_i^n(\mathbb{D}_n) \ge \sum_{i=1}^{r} h_{o^i}^n(\mathbb{D}_n) = \Delta h', \quad (24)$$

which completes the proof.

*Theorem 4. Given a two-layer converged neural network including $m$ neurons with the ReLU activation function. The obtained probability score $p_s(\mathbb{D}_n)$ over the dataset $\mathbb{D}_n$ of some specific client $n$, after running on heterogeneous clients with sub-models extracted through either random or sequential neuron selection strategies, is smaller than our distribution-aware selection method $p_s'(\mathbb{D}_n)$ under the worst-case, i.e., $p_s(\mathbb{D}_n) \le p_s'(\mathbb{D}_n)$.*

*Proof*: We consider there are a total of $m$ neurons in the hidden layer of the global neural network. We assume that neurons numbered 1 to $r$ are the neurons with the highest activation values on client $n_1$ with the dataset $\mathbb{D}_n$ comprising samples belonging to class $s$. In the following, we show that the probability score of the global model over client $n_1$ will be reduced when the neurons are allocated without considering their relationship to data distribution.

Under the worst case, on the $t$-th round, we consider the sub-models extracted by clients $S_t$ with heterogeneous distribution to the client $n_1$ contain neurons numbered 1 to $r$ while the client $n$ does not participate in this round. Based on Theorem 1, the activation values of the local sub-models obtained by these clients are 0 when there are sufficient local training iterations. By denoting the local parameters for the $i$-th neuron in the selected client $n_j$ as $\mathbf{w}_{1,i}^{n_j}$, then we have

$$h_i^{n_j}(\mathbb{D}_n) = \sum_{\mathbf{x}^k \in \mathbb{D}_n} \text{ReLU}((\mathbf{w}_{1,i}^{n_j})^T \mathbf{x}^k) = 0, \quad (25)$$

for each client $n_j \in S_t$ that contains the $i$-th selected neuron. After that, the parameters of each neuron in different clients are aggregated correspondingly in a FedAvg manner, and the global parameters of the $i$-th neuron are $\mathbf{w}_{1,i} = \frac{1}{|S_t|} \sum_{n_j \in S_t} \mathbf{w}_{1,i}^{n_j}$. Now, we can obtain the overall activation value $h_i(\mathbb{D}_n)$ of each $i$ neuron in 1 to $r$ of the global model on the dataset $\mathbb{D}_n$ of the client $n$:

$$h_i(\mathbb{D}_n) = \sum_{\mathbf{x}^k \in \mathbb{D}_n} \text{ReLU}(\mathbf{w}_{1,i}^T \mathbf{x}^k)$$

$$= \sum_{\mathbf{x}^k \in \mathbb{D}_n} \text{ReLU}((\frac{1}{|S_t|} \sum_{n_j \in S_t} \mathbf{w}_{1,i}^{n_j})^T \mathbf{x}^k)$$

$$= \sum_{\mathbf{x}^k \in \mathbb{D}_n} \text{ReLU}\left(\frac{1}{|S_t|} \sum_{n_j \in S_t} (\mathbf{w}_{1,i}^{n_j})^T \mathbf{x}^k\right)$$

$$\underset{(a)}{\leq} \frac{1}{|S_t|} \sum_{\mathbf{x}^k \in \mathbb{D}_n} \sum_{n_j \in S_t} \text{ReLU}((\mathbf{w}_{1,i}^{n_j})^T \mathbf{x}^k)$$

$$= 0, \tag{26}$$

where the inequality (a) is due to the convexity of the ReLU function. Since the activation $h_i(\mathbb{D}_n) \leq 0$, we have $h_i(\mathbb{D}_n) = 0$. As a consequence, the activation of neurons numbered 1 to $r$ in the global model over some specific client $n$ will be significantly reduced with an inappropriate selection strategy.

Since the classifier parameters $w_{2,s,i}$ connected the $i$-th hidden neuron and the class $s$ approach are not selected when $r + 1 \leq i \leq m$, their value approaches the activation mean of samples with the class $s$ when the model converges based on Theorem 1 in [1]. Hence, we denote $w_{2,s,i} = \alpha_s h_i^s$ and $w_{2,c,i} = \alpha_c h_i^c$, for $r + 1 \leq i \leq m$, where $\alpha_s > 0$ and $\alpha_c > 0$ are constants. Then, the probability score $p_s$ of the global model over the dataset $\mathbb{D}_n$ becomes:

$$p_s = \sum_{\mathbf{x}^k \in \mathbb{D}_n} \frac{e^{\mathbf{w}_{2,s} \mathbf{h}^k}}{\sum_{c=1}^{C} e^{\mathbf{w}_{2,c} \mathbf{h}^k}}$$

$$= \sum_{\mathbf{x}^k \in \mathbb{D}_n} \frac{e^{\sum_{i=1}^{r} w_{2,s,i} \cdot 0 + \sum_{i=r+1}^{m} w_{2,s,i} h_i^k}}{e^{\sum_{i=1}^{r} w_{2,s,i} \cdot 0 + \sum_{i=r+1}^{m} w_{2,s,i} h_i^k} + e^{\sum_{i=1}^{r} w_{2,c,i} \cdot 0 + \sum_{i=r+1}^{m} w_{2,c,i} h_i^k}}$$

$$\approx D \frac{e^{\alpha_s \sum_{i=r+1}^{m} (h_i^s)^2}}{e^{\alpha_s \sum_{i=r+1}^{m} (h_i^s)^2} + e^{\alpha_c \sum_{i=r+1}^{m} h_i^c h_i^s}}$$

$$\leq D \frac{e^{\alpha_s \sum_{i=r+1}^{m} (h_i^s)^2}}{e^{\alpha_s \sum_{i=r+1}^{m} (h_i^s)^2} + e^{\alpha_c (m-r) h_{min}^c h_{min}^s}}, \tag{27}$$

where $h_{min}^s$ and $h_{min}^c$ is the minimum activation among all neurons for class $s$ and $c$ respectively.

As our method selects neurons according to the distribution of each client, we contend that the neurons chosen by clients $S_t$ with heterogeneous distributions are not the neurons numbered 1 to $r$ which are top neurons over the client $n$. We consider the neurons selected by clients $S_t$ to be numbered $n^1$ to $n^r$. Similar to (27), we can derive the probability score $p_s'$ of the global model $\mathbf{w}$ over the the training dataset $\mathbb{D}_n$ is

$$p_s' \approx D \frac{e^{\alpha_s \sum_{i=1}^{m-r} (h_{n^i}^s)^2}}{e^{\alpha_s \sum_{i=1}^{m-r} (h_{n^i}^s)^2} + e^{\alpha_c \sum_{i=1}^{m-r} h_{n^i}^c h_{n^i}^s}}$$

$$\leq D \frac{e^{\alpha_s \sum_{i=1}^{m-r} (h_{n^i}^s)^2}}{e^{\alpha_s \sum_{i=1}^{m-r} (h_{n^i}^s)^2} + e^{\alpha_c (m-r) h_{min}^c h_{min}^s}}, \tag{28}$$

Since $h_i^s \leq h_j^s$ for any $1 \leq i \leq r, r + 1 \leq j \leq m$, we have $e^{\alpha_s \sum_{i=r+1}^{m} (h_i^s)^2} \leq e^{\alpha_s \sum_{i=1}^{m-r} (h_{n^i}^s)^2}$. Hence, the upper bound of the probability score $p_s$ is smaller than $p_s'$, i.e.,

$$D \frac{e^{\alpha_s \sum_{i=r+1}^{m} (h_i^s)^2}}{e^{\alpha_s \sum_{i=r+1}^{m} (h_i^s)^2} + e^{\alpha_c (m-r) h_{min}^c h_{min}^s}} \leq D \frac{e^{\alpha_s \sum_{i=1}^{m-r} (h_{n^i}^s)^2}}{e^{\alpha_s \sum_{i=1}^{m-r} (h_{n^i}^s)^2} + e^{\alpha_c (m-r) h_{min}^c h_{min}^s}}, \tag{29}$$

demonstrating our method's effectiveness.

## C    PROOFS OF CONVERGENCE THEORIES

### C.1    General Lemmas

Without losing generality, we in this paper consider that the size of the local dataset in each client is the same and all clients are selected in each round. For ease of analysis, we introduce the index $k$ where $k = t * E + e$. According to Algorithm 1, we have the following basic update formula:

$$\mathbf{w}_{k+1}^n = \mathbf{w}_k^n - \eta \nabla_{\mathbf{w}_k^n} f_n(\mathbf{w}_k^n). \tag{30}$$

We consider the following auxiliary global model $\bar{\mathbf{w}}_{k+1}$, which helps analyze the bound of local updates:

$$\bar{\mathbf{w}}_{k+1} = \bar{\mathbf{w}}_k - \eta \frac{1}{N} \sum_{n=1}^{N} \nabla_{\mathbf{w}_k^n} f_n(\mathbf{w}_k^n) \tag{31}$$

Obviously, $\bar{\mathbf{w}}_k = \mathbf{w}_t$, when $k = t * E$. Besides, We define $\hat{\mathbf{w}}_k^n$ as the full model which fills the sub-model $\mathbf{w}_k^n$ with the global parameters in the latest global round. According to the updating formula, we have $\hat{\mathbf{w}}_{t,e}^n = \mathbf{w}_t$ for all local iteration $e$, and $\frac{1}{N} \sum_{n=1}^{N} \hat{\mathbf{w}}_k^n = \bar{\mathbf{w}}_k$ for all $k$.

Correspondingly, we also introduce an auxiliary full model which helps analyze the bound of the sub-model:

$$\tilde{\mathbf{w}}_{k+1} = \tilde{\mathbf{w}}_k - \eta \frac{1}{N} \sum_{n=1}^{N} \nabla_{\hat{\mathbf{w}}_k^n} f_n(\hat{\mathbf{w}}_k^n). \tag{32}$$

*Lemma 1'. Consider the gradient $\nabla_{\mathbf{w} \odot \mathbf{M}} f(\mathbf{w} \odot \mathbf{M})$ calculated from a sub-model $\mathbf{w} \odot \mathbf{M}$ and another gradient $Q(\nabla_{\mathbf{w}} f(\mathbf{w})) = \nabla_{\mathbf{w}}(\mathbf{h}_m) \nabla_{\mathbf{h}_m} f(\mathbf{h}_n)$ computed from the entire model $\mathbf{w}$ but with the activation of neurons pruned by the sub-model set to zero, i.e., $\mathbf{h}_m = \mathbf{h} \odot \mathbf{m}$. For neural networks that use the ReLU activation function, these two gradients are equivalent, meaning that $\nabla_{\mathbf{w} \odot \mathbf{M}} f(\mathbf{w} \odot \mathbf{M}) = Q(\nabla_{\mathbf{w}} f(\mathbf{w}))$.*

PROOF. We prove this lemma by showing that pruning neurons is equivalent to setting the activation of these neurons to be zero in both the forward and backward process. Considering the $p$-th neuron in the $(l-1)$-th layer is pruned, then the activation of each $i$-th neuron in the $l$-th layer is

$$h_{l,i} = \sigma \Big( \sum_{j=1, j \neq p}^{m_{l-1}} w_{l,i,j} h_{l-1,j} + b_{l,i} \Big), \tag{33}$$

which is equivalent to setting $h_{l-1,p} = 0$.

We now prove that the gradient is equivalent based on the backward process of gradient computing. The parameters not connected to pruned neurons are nothing related to their activation in the gradient computation process, which naturally remains the same. Considering this, we mainly focus on the parameters connected to the pruned neurons. Since the gradients of these parameters connected to the pruned neurons are zero, we can prove this conclusion by showing that the gradients of parameters connected to the neurons with zero activation are also zero. Specifically, we divide the parameters connected to the neuron into two types, inputting parameters and outputting parameters according to their relative position to the given neuron. Define the non-activated feature as

$$a_{l,i} = \sum_{j=1}^{m_{l-1}} w_{l,i,j} h_{l-1,j} + b_{l,i} \tag{34}$$

and the error received back from the $p$-th neuron in $l+1$-th layer as $\delta_{l+1,p}$. The gradient $\nabla_{w_{l,i,j}} f(\mathbf{w})$ of each outputting parameter for the $j$-th neuron in the $(l-1)$-th layer is

$$\nabla_{w_{l,i,j}} f(\mathbf{w}) = h_{l-1,j} \nabla_{a_{l,i}} h_{l,i}(a_{l,i}) \sum_{p=1}^{m_{l+1}} w_{l+1,p,i} \delta_{l+1,p}. \tag{35}$$

Obviously, by setting the activation $h_{l-1,j}$ to be zero, its outputting parameters also become zero, which equals to pruning the neuron. Since $\nabla_{a_{l-1,j}} h_{l-1,j}(a_{l-1,j}) = 0$ holds for each neuron with the ReLU activation function, the gradients of its connected inputting parameters are

$$\nabla_{w_{l-1,j,q}} f(\mathbf{w}) = h_{l-2,q} \nabla_{a_{l-1,j}} h_{l-1,j}(a_{l-1,j}) \sum_{i=1}^{m_l} w_{l,i,j} \delta_{l,i} = h_{l-2,q} \cdot 0 \cdot \sum_{i=1}^{m_l} w_{l,i,j} \delta_{l,i} = 0, \tag{36}$$

which completes the proof. □

*Lemma 1 The error of the gradient calculated by the sub-model is bounded by*

$$\mathbb{E} \| \nabla_{\hat{\mathbf{w}}_k^n} f_n(\hat{\mathbf{w}}_k^n) - \nabla_{\mathbf{w}_k^n} f_n(\mathbf{w}_k^n) \|_2^2 \leq G_h^2 H^2 \sum_{l=1}^{L} \mathbb{E} \sum_{i \in S_{l,k}} \| \mathbf{e}_{l,i,k} \|_2^2 + \sum_{l=1}^{L} \mathbb{E} \sum_{i \in S_{l,k}^c} \| \nabla_{\hat{\mathbf{w}}_{l,i,k}^n} f_n(\hat{\mathbf{w}}_k^n) \|_2^2, \tag{37}$$

*where $S_{l,\tau}$ is the set of selected neurons in the $l$-th layer and $S_{l,\tau}^c$ denotes the set of un-selected neurons. $\hat{\mathbf{w}}_{l,i,k-1}^n$ represents the parameters connected to the neuron $i$.*

PROOF. : $\mathbb{E} \| \nabla_{\hat{\mathbf{w}}_{k-1}^n} f_n(\hat{\mathbf{w}}_{k-1}^n) - \nabla_{\mathbf{w}_{k-1}^n} f_n(\mathbf{w}_{k-1}^n) \|_2^2$ measures the distance between the gradient computed from the full model and from the sub-model. To calculate this distance, we use Lemma **??** to transform the gradient that was computed from the sub-model into the gradient of the entire model.

Specifically, according to the chain rule of backward, the gradient of the parameters of $i$-th neuron in $l$-th layer for the entire model $\hat{\mathbf{w}}_{k-1}^n$ is $\nabla_{\hat{\mathbf{w}}_{l,i,k-1}^n} \mathbf{h}_{l,i,k-1}^n \nabla_{\mathbf{h}_{l,i,k-1}^n} f_n(\mathbf{h}_{k-1}^n)$. Similarly, the gradient of the parameters connected to the $i$-th non-pruned neuron in the $l$-th layer of the sub-model $\mathbf{w}_{k-1}^n = \hat{\mathbf{w}}_{k-1}^n \odot \mathbf{M}_{k-1}^n$ is $\nabla_{\hat{\mathbf{w}}_{l,i,k-1}^n} \mathbf{h}_{l,i,k-1}^n \nabla_{\mathbf{h}_{l,i,k-1}^n} f_n(\mathbf{h}_{m,k-1}^n)$ where $\mathbf{h}_{m,k-1} = \mathbf{h}_{k-1} \odot \mathbf{m}_{k-1}$. We define error between them is $\mathbf{e}_{k-1}^n = \mathbf{h}_{m,k-1}^n - \mathbf{h}_{k-1}^n$. We use $S_{l,k-1}$ to denote the set of selected neurons in the $l$-th layer and $S_{l,k-1}^c$ to denote its complementary set in the $l$-th layer, i.e., the set of unselected neurons. We utilize Taylor expansion to $\nabla_{\mathbf{h}_{l,i,k-1}^n} f_n(\mathbf{h}_{m,k-1}^n)$ around the full activation point $\mathbf{h}_{k-1}^n$, obtaining:

$$\nabla_{\mathbf{h}_{l,i,k-1}^n} f_n(\mathbf{h}_{m,k-1}^n) = \nabla_{\mathbf{h}_{l,i,k-1}^n} f_n(\mathbf{h}_{k-1}^n) + R(\mathbf{e}_{k-1}^n)$$

$$= \nabla_{\mathbf{h}_{l,i,k-1}^n} f_n(\mathbf{h}_{k-1}^n) + \nabla_{\mathbf{h}_{l,i,k-1}^n}^2 f_n(\mathbf{h}_{k-1}^n)^T \mathbf{e}_{l,i,k-1} + \dots, \tag{38}$$

where $R(\mathbf{e}_{l,i,k-1}^n)$ denotes the infinite sum of all terms from the second partial derivatives. Based on the Assumption 3 and basics of the Taylor series, we obtain the approximation error:

$$\|R(\mathbf{e}_{l,i,k-1}^n)\|_2^2 \leq H^2 \|\mathbf{e}_{l,i,k-1}\|_2^2. \tag{39}$$

Then, we have the following inequality:

$$\mathbb{E}\|\nabla_{\hat{\mathbf{w}}_{k-1}^n} f_n(\hat{\mathbf{w}}_{k-1}^n) - \nabla_{\mathbf{w}_{k-1}^n} f_n(\mathbf{w}_{k-1}^n)\|_2^2$$

$$= \sum_{l=1}^{L} \mathbb{E} \sum_{i \in S_{l,k-1}} \|\nabla_{\hat{\mathbf{w}}_{l,i,k-1}^n} \mathbf{h}_{l,i,k-1}^n \nabla_{\mathbf{h}_{l,i,k-1}^n} f_n(\mathbf{h}_{k-1}^n) - \nabla_{\hat{\mathbf{w}}_{l,i,k-1}^n} \mathbf{h}_{l,i,k-1}^n \nabla_{\mathbf{h}_{l,i,k-1}^n} f_n(\mathbf{h}_{m,k-1}^n)\|_2^2$$

$$+ \sum_{l=1}^{L} \mathbb{E} \sum_{i \in S_{l,k-1}^c} \|\nabla_{\hat{\mathbf{w}}_{l,i,k-1}^n} \mathbf{h}_{l,i,k-1}^n \nabla_{\mathbf{h}_{l,i,k-1}^n} f_n(\mathbf{h}_{k-1}^n)\|_2^2$$

$$= \sum_{l=1}^{L} \mathbb{E} \sum_{i \in S_{l,k-1}} \|\nabla_{\hat{\mathbf{w}}_{l,i,k-1}^n} \mathbf{h}_{l,i,k-1}^n \nabla_{\mathbf{h}_{l,i,k-1}^n} f_n(\mathbf{h}_{k-1}^n) - \nabla_{\hat{\mathbf{w}}_{l,i,k-1}^n} \mathbf{h}_{l,i,k-1}^n (\nabla_{\mathbf{h}_{l,i,k-1}^n} f_n(\mathbf{h}_{k-1}^n) + R(\mathbf{e}_{k-1}^n))\|_2^2$$

$$+ \sum_{l=1}^{L} \mathbb{E} \sum_{i \in S_{l,k}^c} \|\nabla_{\hat{\mathbf{w}}_{l,i,k}^n} f_n(\hat{\mathbf{w}}_k^n)\|_2^2$$

$$= \sum_{l=1}^{L} \mathbb{E} \sum_{i \in S_{l,k-1}} \|\nabla_{\hat{\mathbf{w}}_{l,i,k-1}^n} \mathbf{h}_{l,i,k-1}^n R(\mathbf{e}_{k-1}^n)\|_2^2 + \sum_{l=1}^{L} \mathbb{E} \sum_{i \in S_{l,k}^c} \|\nabla_{\hat{\mathbf{w}}_{l,i,k}^n} f_n(\hat{\mathbf{w}}_k^n)\|_2^2,$$

$$\underset{(a)}{\leq} \sum_{l=1}^{L} \mathbb{E} \sum_{i \in S_{l,k-1}} G_h^2 H^2 \|\mathbf{e}_{l,i,k-1}\|_2^2 + \sum_{l=1}^{L} \mathbb{E} \sum_{i \in S_{l,k}^c} \|\nabla_{\hat{\mathbf{w}}_{l,i,k}^n} f_n(\hat{\mathbf{w}}_k^n)\|_2^2, \tag{40}$$

where $(a)$ follows from Assumption 4. The proof is done. □

## C.2  Proof of Theorem 5

*Theorem 5. Considering $F_*$ be the global minima of the loss function, $\gamma$ and $\alpha$ are constants with $\gamma > 0$, $0 \leq \alpha < 1$, and the learning rate $0 < \eta \leq \frac{1}{L_s}$, then for all neural networks with ReLU activation function, the expected average of the squared gradient norms of $F$ obtained by Algorithm 1 satisfies the following bound for all $K \in \mathbb{N}$:*

$$\sum_{k=1}^{K} \mathbb{E}\|\frac{1}{N} \sum_{n=1}^{N} \nabla_{\hat{\mathbf{w}}_k^n} F_n(\hat{\mathbf{w}}_k^n)\|_2^2 \leq \frac{2(F(\mathbf{w}_1) - F_*)}{\eta} + 2L_s^2 \eta^2 \alpha G^2 (1+\gamma) \frac{(1+\gamma)^{K-1} - 1}{\gamma^2} + \frac{KL_s \eta \sigma_2^2}{N}$$

$$+ 4L_s^2 \eta^2 (1 + \frac{1}{\gamma}) G_h^2 H^2 \sum_{k=1}^{K} \sum_{\tau=1}^{k-1} (1+\gamma)^{k-1-\tau} \sum_{l=1}^{L} \mathbb{E} \sum_{i \in S_{l,\tau}} \|\mathbf{e}_{l,i,\tau}\|_2^2 + 16KL_s^2 \eta^4 E^2 G^2 (1 + \frac{1}{\gamma}), \tag{41}$$

*where $S_{l,\tau}$ is the set of selected neurons in the l-th layer.*

PROOF.  : Our proof starts from the L-smooth assumption (Assumption 1) that bounds the loss of one global iteration:

$$\mathbb{E}(F(\tilde{\mathbf{w}}_{k+1}) - F(\tilde{\mathbf{w}}_k)) \leq \mathbb{E}\langle \nabla F(\tilde{\mathbf{w}}_k), \tilde{\mathbf{w}}_{k+1} - \tilde{\mathbf{w}}_k \rangle + \frac{L_s}{2} \mathbb{E}\|\tilde{\mathbf{w}}_{k+1} - \tilde{\mathbf{w}}_k\|_2^2. \tag{42}$$

The inequality contains two items and we bound them separately in the following text:

$$\mathbb{E}\|\tilde{\mathbf{w}}_{k+1} - \tilde{\mathbf{w}}_k\|_2^2 = \eta^2 \mathbb{E}\|\frac{1}{N} \sum_{n=1}^{N} \nabla_{\hat{\mathbf{w}}_k^n} f_n(\hat{\mathbf{w}}_k^n)\|_2^2$$

$$\underset{(a)}{=} \eta^2 \mathbb{E}\|\frac{1}{N} \sum_{n=1}^{N} (\nabla_{\hat{\mathbf{w}}_k^n} f_n(\hat{\mathbf{w}}_k^n) - \nabla_{\hat{\mathbf{w}}_k^n} F_n(\hat{\mathbf{w}}_k^n))\|_2^2 + \eta^2 \mathbb{E}\|\frac{1}{N} \sum_{n=1}^{N} \nabla_{\hat{\mathbf{w}}_k^n} F_n(\hat{\mathbf{w}}_k^n)\|_2^2$$

$$\underset{(b)}{=} \eta^2 \frac{1}{N^2} \sum_{n=1}^{N} \mathbb{E}\|\nabla_{\hat{\mathbf{w}}_k^n} f_n(\hat{\mathbf{w}}_k^n) - \nabla_{\hat{\mathbf{w}}_k^n} F_n(\hat{\mathbf{w}}_k^n)\|_2^2 + \eta^2 \mathbb{E}\|\frac{1}{N} \sum_{n=1}^{N} \nabla_{\hat{\mathbf{w}}_k^n} F_n(\hat{\mathbf{w}}_k^n)\|_2^2$$

$$\underset{(c)}{\leq} \frac{1}{N} \eta^2 \sigma_2^2 + \eta^2 \mathbb{E}\|\frac{1}{N} \sum_{n=1}^{N} \nabla_{\hat{\mathbf{w}}_k^n} F_n(\hat{\mathbf{w}}_k^n)\|_2^2, \tag{43}$$

where $(b)$ follows that $\mathbb{E}\nabla_{\hat{\mathbf{w}}_k^n} f_n(\hat{\mathbf{w}}_k^n) = \mathbb{E}\nabla_{\hat{\mathbf{w}}_k^n} F_n(\hat{\mathbf{w}}_k^n)$ and $\mathbb{E}\|\mathbf{v}\|^2 = \mathbb{E}\|\mathbf{v} - \mathbb{E}\mathbf{v}\|^2 + \|\mathbb{E}\mathbf{v}\|^2$. $(b)$ is due to the independence among clients and the zero mean and $(c)$ follows from the Assumption 2.

For another item, we have

$$\mathbb{E}\left\langle \nabla F(\tilde{\mathbf{w}}_k), \tilde{\mathbf{w}}_{k+1} - \tilde{\mathbf{w}}_k \right\rangle = \mathbb{E}\left\langle \nabla F(\tilde{\mathbf{w}}_k), -\eta \frac{1}{N} \sum_{n=1}^{N} \nabla_{\hat{\mathbf{w}}_k^n} f_n(\hat{\mathbf{w}}_k^n) \right\rangle$$

$$= -\eta \mathbb{E}\left\langle \nabla F(\tilde{\mathbf{w}}_k), \frac{1}{N} \sum_{n=1}^{N} \nabla_{\hat{\mathbf{w}}_k^n} F_n(\hat{\mathbf{w}}_k^n) \right\rangle$$

$$= -\frac{\eta}{2}\left[ \mathbb{E}\|\nabla F(\tilde{\mathbf{w}}_k)\|_2^2 + \mathbb{E}\|\frac{1}{N} \sum_{n=1}^{N} \nabla_{\hat{\mathbf{w}}_k^n} F_n(\hat{\mathbf{w}}_k^n)\|_2^2 - \mathbb{E}\|\nabla F(\tilde{\mathbf{w}}_k) - \frac{1}{N} \sum_{n=1}^{N} \nabla_{\hat{\mathbf{w}}_k^n} F_n(\hat{\mathbf{w}}_k^n)\|_2^2 \right]. \tag{44}$$

Substituting (43) and (44) into (42) derives

$$\mathbb{E}(F(\tilde{\mathbf{w}}_{k+1}) - F(\tilde{\mathbf{w}}_k)) \le -\frac{\eta}{2}\mathbb{E}\|\nabla F(\tilde{\mathbf{w}}_k)\|_2^2 - \frac{\eta - \eta^2 L_s}{2}\mathbb{E}\|\frac{1}{N} \sum_{n=1}^{N} \nabla_{\hat{\mathbf{w}}_k^n} F_n(\hat{\mathbf{w}}_k^n)\|_2^2$$

$$+ \frac{\eta}{2}\mathbb{E}\|\nabla F(\tilde{\mathbf{w}}_k) - \frac{1}{N} \sum_{n=1}^{N} \nabla_{\hat{\mathbf{w}}_k^n} F_n(\hat{\mathbf{w}}_k^n)\|_2^2 + \frac{L_s \eta^2 \sigma_2^2}{2N}$$

$$\underset{(a)}{\le} -\frac{\eta}{2}\mathbb{E}\|\nabla F(\tilde{\mathbf{w}}_k)\|_2^2 + \frac{\eta}{2}\mathbb{E}\|\nabla F(\tilde{\mathbf{w}}_k) - \frac{1}{N} \sum_{n=1}^{N} \nabla_{\hat{\mathbf{w}}_k^n} F_n(\hat{\mathbf{w}}_k^n)\|_2^2 + \frac{L_s \eta^2 \sigma_2^2}{2N},$$

$$\le -\frac{\eta}{2}\left( \mathbb{E}\|\nabla F(\tilde{\mathbf{w}}_k)\|_2^2 + \mathbb{E}\|\nabla F(\tilde{\mathbf{w}}_k) - \frac{1}{N} \sum_{n=1}^{N} \nabla_{\hat{\mathbf{w}}_k^n} F_n(\hat{\mathbf{w}}_k^n)\|_2^2 \right)$$

$$+ \eta \mathbb{E}\|\nabla F(\tilde{\mathbf{w}}_k) - \frac{1}{N} \sum_{n=1}^{N} \nabla_{\hat{\mathbf{w}}_k^n} F_n(\hat{\mathbf{w}}_k^n)\|_2^2 + \frac{L_s \eta^2 \sigma_2^2}{2N}, \tag{45}$$

where $(a)$ holds when $0 < \eta < \frac{1}{L}$. Since the following inequality holds:

$$\mathbb{E}\|\frac{1}{N} \sum_{n=1}^{N} \nabla_{\hat{\mathbf{w}}_k^n} F_n(\hat{\mathbf{w}}_k^n)\|_2^2$$

$$= \mathbb{E}\|\nabla F(\tilde{\mathbf{w}}_k) - \left(\nabla F(\tilde{\mathbf{w}}_k) - \frac{1}{N} \sum_{n=1}^{N} \nabla_{\hat{\mathbf{w}}_k^n} F_n(\hat{\mathbf{w}}_k^n)\right)\|_2^2$$

$$\le \mathbb{E}\|\nabla F(\tilde{\mathbf{w}}_k)\|_2^2 + \mathbb{E}\|\nabla F(\tilde{\mathbf{w}}_k) - \frac{1}{N} \sum_{n=1}^{N} \nabla_{\hat{\mathbf{w}}_k^n} F_n(\hat{\mathbf{w}}_k^n)\|_2^2, \tag{46}$$

we have the following inequality by re-organizing (45):

$$\frac{\eta}{2}\mathbb{E}\|\frac{1}{N} \sum_{n=1}^{N} \nabla_{\hat{\mathbf{w}}_k^n} F_n(\hat{\mathbf{w}}_k^n)\|_2^2 + \mathbb{E}(F(\tilde{\mathbf{w}}_{k+1}) - F(\tilde{\mathbf{w}}_k))$$

$$\le \eta \mathbb{E}\|\nabla F(\tilde{\mathbf{w}}_k) - \frac{1}{N} \sum_{n=1}^{N} \nabla_{\hat{\mathbf{w}}_k^n} F_n(\hat{\mathbf{w}}_k^n)\|_2^2 + \frac{L_s \eta^2 \sigma_2^2}{2N}. \tag{47}$$

We denote the minimum of the loss function by $F_*$. By computing the sum of (47) from $k = 1$ to $K$, we can obtain:

$$\sum_{k=1}^{K} \frac{\eta}{2}\mathbb{E}\|\frac{1}{N} \sum_{n=1}^{N} \nabla_{\hat{\mathbf{w}}_k^n} F_n(\hat{\mathbf{w}}_k^n)\|_2^2 + (F_* - F(\tilde{\mathbf{w}}_1))$$

$$\le \sum_{k=1}^{K} \frac{\eta}{2}\mathbb{E}\|\frac{1}{N} \sum_{n=1}^{N} \nabla_{\hat{\mathbf{w}}_k^n} F_n(\hat{\mathbf{w}}_k^n)\|_2^2 + \mathbb{E}(F(\tilde{\mathbf{w}}_{K+1}) - F(\tilde{\mathbf{w}}_1))$$

$$\le \eta \sum_{k=1}^{K} \mathbb{E}\|\nabla F(\tilde{\mathbf{w}}_k) - \frac{1}{N} \sum_{n=1}^{N} \nabla_{\hat{\mathbf{w}}_k^n} F_n(\hat{\mathbf{w}}_k^n)\|_2^2 + \frac{K L_s \eta^2 \sigma_2^2}{2N}. \tag{48}$$

Now, we seek to present the bound of $\mathbb{E}\|\nabla F(\tilde{\mathbf{w}}_k) - \frac{1}{N} \sum_{n=1}^{N} \nabla_{\hat{\mathbf{w}}_k^n} F_n(\hat{\mathbf{w}}_k^n)\|_2^2$:

$$\mathbb{E}\|\nabla F(\tilde{\mathbf{w}}_k) - \frac{1}{N} \sum_{n=1}^{N} \nabla_{\hat{\mathbf{w}}_k^n} F_n(\hat{\mathbf{w}}_k^n)\|_2^2$$

$$= \mathbb{E}\|\frac{1}{N} \sum_{n=1}^{N} \nabla F_n(\tilde{\mathbf{w}}_k) - \frac{1}{N} \sum_{n=1}^{N} \nabla_{\hat{\mathbf{w}}_k^n} F_n(\hat{\mathbf{w}}_k^n)\|_2^2$$

$$= \frac{1}{N^2} \mathbb{E} \| \sum_{n=1}^{N} \nabla F_n(\tilde{\mathbf{w}}_k) - \sum_{n=1}^{N} \nabla_{\hat{\mathbf{w}}_k^n} F_n(\hat{\mathbf{w}}_k^n) \|_2^2$$

$$\leq \frac{1}{N} \sum_{n=1}^{N} \mathbb{E} \| \nabla F_n(\tilde{\mathbf{w}}_k) - \nabla_{\hat{\mathbf{w}}_k^n} F_n(\hat{\mathbf{w}}_k^n) \|_2^2$$

$$\underset{(a)}{\leq} \frac{L_s^2}{N} \sum_{n=1}^{N} \mathbb{E} \| \tilde{\mathbf{w}}_k - \hat{\mathbf{w}}_k^n \|_2^2$$

$$\leq \frac{L_s^2}{N} \sum_{n=1}^{N} \mathbb{E} \| \tilde{\mathbf{w}}_k - \bar{\mathbf{w}}_k + \bar{\mathbf{w}}_k - \hat{\mathbf{w}}_k^n \|_2^2,$$

$$\leq \frac{2L_s^2}{N} \sum_{n=1}^{N} \left( \mathbb{E} \| \tilde{\mathbf{w}}_k - \bar{\mathbf{w}}_k \|_2^2 + \mathbb{E} \| \bar{\mathbf{w}}_k - \hat{\mathbf{w}}_k^n \|_2^2 \right), \tag{49}$$

where (a) follows from Assumption 1. The item $\| \tilde{\mathbf{w}}_k - \bar{\mathbf{w}}_k \|_2^2$ represents the error between the ideal model updated based on the entire model and the real model updated based on the sub-model. The item $\| \bar{\mathbf{w}}_k - \hat{\mathbf{w}}_k^n \|_2^2$ represents the error between the global model and the local model. Next, we bound them separately.

First, considering the previous synchronization iteration is $k_0$, we have

$$\mathbb{E} \| \bar{\mathbf{w}}_k - \hat{\mathbf{w}}_k^n \|_2^2$$

$$= \mathbb{E} \| (\bar{\mathbf{w}}_{k_0} - \eta \sum_{\tau=k_0}^{k} \frac{1}{N} \sum_{n=1}^{N} \nabla_{\mathbf{w}_\tau^n} f_n(\mathbf{w}_\tau^n)) - (\hat{\mathbf{w}}_{k_0}^n - \eta \sum_{\tau=k_0}^{k} \nabla_{\mathbf{w}_\tau^n} f_n(\mathbf{w}_\tau^n)) \|_2^2$$

$$\underset{(a)}{=} \eta^2 \mathbb{E} \| \sum_{\tau=k_0}^{k} \frac{1}{N} \sum_{n=1}^{N} \nabla_{\mathbf{w}_\tau^n} f_n(\mathbf{w}_\tau^n) - \sum_{\tau=k_0}^{k} \nabla_{\mathbf{w}_\tau^n} f_n(\mathbf{w}_\tau^n) \|_2^2$$

$$\underset{(b)}{\leq} 2\eta^2 \mathbb{E} \| \sum_{\tau=k_0}^{k} \frac{1}{N} \sum_{n=1}^{N} \nabla_{\mathbf{w}_\tau^n} f_n(\mathbf{w}_\tau^n) \|_2^2 + 2\eta^2 \mathbb{E} \| \sum_{\tau=k_0}^{k} \nabla_{\mathbf{w}_\tau^n} f_n(\mathbf{w}_\tau^n) \|_2^2$$

$$\underset{(c)}{\leq} 2\eta^2 (k - k_0) \sum_{\tau=k_0}^{k} \mathbb{E} \| \frac{1}{N} \sum_{n=1}^{N} \nabla_{\mathbf{w}_\tau^n} f_n(\mathbf{w}_\tau^n) \|_2^2 + 2\eta^2 (k - k_0) \sum_{\tau=k_0}^{k} \mathbb{E} \| \nabla_{\mathbf{w}_\tau^n} f_n(\mathbf{w}_\tau^n) \|_2^2$$

$$\underset{(d)}{\leq} 2\eta^2 (k - k_0) \sum_{\tau=k_0}^{k} \frac{1}{N} \sum_{n=1}^{N} \mathbb{E} \| \nabla_{\mathbf{w}_\tau^n} f_n(\mathbf{w}_\tau^n) \|_2^2 + 2\eta^2 (k - k_0) \sum_{\tau=k_0}^{k} \mathbb{E} \| \nabla_{\mathbf{w}_\tau^n} f_n(\mathbf{w}_\tau^n) \|_2^2$$

$$\underset{(e)}{\leq} 4\eta^2 E^2 G^2, \tag{50}$$

where $(a)$ holds because $\bar{\mathbf{w}}_{k_0} = \hat{\mathbf{w}}_{k_0}^n = \mathbf{w}_{k_0}$. $(c) - (d)$ come from the Cauchy-Schwarz Inequality. $(e)$ is due to Assumption 2.

For another item, we have

$$\mathbb{E} \| \tilde{\mathbf{w}}_k - \bar{\mathbf{w}}_k \|_2^2$$

$$= \mathbb{E} \| (\tilde{\mathbf{w}}_{k-1} - \eta \frac{1}{N} \sum_{n=1}^{N} \nabla_{\hat{\mathbf{w}}_{k-1}^n} f_n(\hat{\mathbf{w}}_{k-1}^n)) - (\bar{\mathbf{w}}_{k-1} - \eta \frac{1}{N} \sum_{n=1}^{N} \nabla_{\mathbf{w}_{k-1}^n} f_n(\mathbf{w}_{k-1}^n)) \|_2^2$$

$$\underset{(a)}{=} (1 + \gamma) \mathbb{E} \| \tilde{\mathbf{w}}_{k-1} - \bar{\mathbf{w}}_{k-1} \|_2^2 + \eta^2 (1 + \frac{1}{\gamma}) \mathbb{E} \| \frac{1}{N} \sum_{n=1}^{N} \nabla_{\hat{\mathbf{w}}_{k-1}^n} f_n(\hat{\mathbf{w}}_{k-1}^n) - \frac{1}{N} \sum_{n=1}^{N} \nabla_{\mathbf{w}_{k-1}^n} f_n(\mathbf{w}_{k-1}^n) \|_2^2$$

$$\leq (1 + \gamma) \mathbb{E} \| \tilde{\mathbf{w}}_{k-1} - \bar{\mathbf{w}}_{k-1} \|_2^2 + \eta^2 (1 + \frac{1}{\gamma}) \frac{1}{N} \sum_{n=1}^{N} \mathbb{E} \| \nabla_{\hat{\mathbf{w}}_{k-1}^n} f_n(\hat{\mathbf{w}}_{k-1}^n) - \nabla_{\mathbf{w}_{k-1}^n} f_n(\mathbf{w}_{k-1}^n) \|_2^2$$

$$= (1 + \gamma) \mathbb{E} \| \tilde{\mathbf{w}}_{k-1} - \bar{\mathbf{w}}_{k-1} \|_2^2 + \eta^2 (1 + \frac{1}{\gamma}) \frac{1}{N} \sum_{n=1}^{N} \mathbb{E} \| \nabla_{\hat{\mathbf{w}}_{k-1}^n} f_n(\hat{\mathbf{w}}_{k-1}^n) - \nabla_{\mathbf{w}_{k-1}^n} f_n(\mathbf{w}_{k-1}^n) \|_2^2$$

$$= \sum_{\tau=1}^{k-1} \eta^2 (1 + \gamma)^{k-1-\tau} (1 + \frac{1}{\gamma}) \frac{1}{N} \sum_{n=1}^{N} \mathbb{E} \| \nabla_{\hat{\mathbf{w}}_\tau^n} f_n(\hat{\mathbf{w}}_\tau^n) - \nabla_{\mathbf{w}_\tau^n} f_n(\mathbf{w}_\tau^n) \|_2^2, \tag{51}$$

where $(a)$ arises from the inequality $(\mathbf{v}_1 + \mathbf{v}_2)^2 \leq (1 + \gamma) \mathbf{v}_1^2 + (1 + \frac{1}{\gamma}) \mathbf{v}_2^2$ for $\gamma > 0$.

Based on Assumption 4, obviously, there is a constant $0 < \alpha < 1$ that

$$\sum_{l=1}^{L} \mathbb{E} \sum_{i \in S_{l,k-1}^c} \|\nabla_{\hat{\mathbf{w}}_{l,i,k-1}^n} f_n(\hat{\mathbf{w}}_k^n)\|_2^2$$

$$\leq \alpha \sum_{l=1}^{L} \sum_{i=1}^{m_l} \mathbb{E} \|\nabla_{\hat{\mathbf{w}}_{l,i,k-1}^n} f_n(\hat{\mathbf{w}}_{k-1}^n)\|_2^2$$

$$= \alpha \mathbb{E} \|\nabla_{\hat{\mathbf{w}}_{k-1}^n} f_n(\hat{\mathbf{w}}_{k-1}^n)\|_2^2$$

$$\leq \alpha G^2. \tag{52}$$

Then, according to Lemma 1, we have

$$\mathbb{E} \|\nabla_{\hat{\mathbf{w}}_{k-1}^n} f_n(\hat{\mathbf{w}}_{k-1}^n) - \nabla_{\mathbf{w}_{k-1}^n} f_n(\mathbf{w}_{k-1}^n)\|_2^2$$

$$\underset{(a)}{\leq} \sum_{l=1}^{L} \mathbb{E} \sum_{i \in S_{l,k-1}} G_h^2 H^2 \|\mathbf{e}_{l,i,k-1}\|_2^2 + \alpha G^2. \tag{53}$$

Bringing (53) back to (51) derives:

$$\mathbb{E} \|\tilde{\mathbf{w}}_k - \bar{\mathbf{w}}_k\|_2^2 \leq \sum_{\tau=1}^{k-1} \eta^2 (1+\gamma)^{k-1-\tau} \left(1+\frac{1}{\gamma}\right) \left( \sum_{l=1}^{L} \mathbb{E} \sum_{i \in S_{l,\tau}} G_h^2 H^2 \|\mathbf{e}_{l,i,\tau}\|_2^2 + \alpha G^2 \right). \tag{54}$$

Substituting (54) and (50) back into (49), and then bringing the derived inequality back into (48) obtains:

$$\sum_{k=1}^{K} \frac{\eta}{2} \mathbb{E} \|\frac{1}{N} \sum_{n=1}^{N} \nabla_{\hat{\mathbf{w}}_k^n} F_n(\hat{\mathbf{w}}_k^n)\|_2^2 + (F_* - F(\tilde{\mathbf{w}}_1))$$

$$\leq 2 L_s^2 \eta^3 \left(1+\frac{1}{\gamma}\right) G_h^2 H^2 \sum_{k=1}^{K} \sum_{\tau=1}^{k-1} (1+\gamma)^{k-1-\tau} \sum_{l=1}^{L} \mathbb{E} \sum_{i \in S_{l,\tau}} \|\mathbf{e}_{l,i,\tau}\|_2^2$$

$$+ L_s^2 \eta^3 \alpha G^2 (1+\gamma) \frac{(1+\gamma)^{K-1}}{\gamma^2} + 8 K L_s^2 \eta^5 E^2 G^2 \left(1+\frac{1}{\gamma}\right) + \frac{K L_s \eta^2 \sigma_2^2}{2N}. \tag{55}$$

Reorganizing the inequality proves the theorem. □

## C.3 Proof of Theorem 6

**Theorem 6.** *Considering $F_*$ be the global minima of the loss function and the learning rate $0 < \eta \leq \frac{1}{4L_s}$, then for all neural networks with ReLU activation function, the expected average of the squared gradient norms of $F$ obtained by Algorithm 1 satisfies the following bound for all $t \in \mathbb{N}$:*

$$\frac{1}{T} \sum_{t=1}^{T} \|\nabla_{\mathbf{w}_t} F(\mathbf{w}_t)\|_2^2 \leq \frac{4(F(\mathbf{w}_1) - F_*)}{\sqrt{T}} + 4E \left(\frac{L_s}{\sqrt{T}} + \frac{1}{2}\right) (G_h^2 H^2 r M \epsilon^2 + \alpha G^2) + \frac{8 E L_s E^2 G^2}{T}, \tag{56}$$

*where $\alpha$ is a constant relying on the extraction ratio of the sub-model with $0 \leq \alpha < 1$.*

PROOF. : Our proof also starts from the L-smooth assumption (Assumption 1) that bounds the loss of one global iteration:

$$\mathbb{E}(F(\bar{\mathbf{w}}_{k+1}) - F(\bar{\mathbf{w}}_k))$$

$$\leq \mathbb{E} \langle \nabla F(\bar{\mathbf{w}}_k), \bar{\mathbf{w}}_{k+1} - \bar{\mathbf{w}}_k \rangle + \frac{L_s}{2} \mathbb{E} \|\bar{\mathbf{w}}_{k+1} - \bar{\mathbf{w}}_k\|_2^2$$

$$= \mathbb{E} \left\langle \nabla F(\bar{\mathbf{w}}_k), -\eta \frac{1}{N} \sum_{n=1}^{N} \nabla_{\mathbf{w}_k^n} F_n(\mathbf{w}_k^n) \right\rangle + \frac{L_s}{2} \mathbb{E} \| -\eta \frac{1}{N} \sum_{n=1}^{N} \nabla_{\mathbf{w}_k^n} F_n(\mathbf{w}_k^n)\|_2^2$$

$$= \eta \mathbb{E} \left\langle \nabla F(\bar{\mathbf{w}}_k), -\frac{1}{N} \sum_{n=1}^{N} \nabla_{\mathbf{w}_k^n} F_n(\mathbf{w}_k^n) + \frac{1}{N} \sum_{n=1}^{N} \nabla_{\hat{\mathbf{w}}_k^n} F_n(\hat{\mathbf{w}}_k^n) \right\rangle + \eta \mathbb{E} \left\langle \nabla F(\bar{\mathbf{w}}_k), -\frac{1}{N} \sum_{n=1}^{N} \nabla_{\hat{\mathbf{w}}_k^n} F_n(\hat{\mathbf{w}}_k^n) \right\rangle$$

$$+ \frac{L_s \eta^2}{2} \mathbb{E} \|\frac{1}{N} \sum_{n=1}^{N} \nabla_{\mathbf{w}_k^n} F_n(\mathbf{w}_k^n) - \frac{1}{N} \sum_{n=1}^{N} \nabla_{\hat{\mathbf{w}}_k^n} F_n(\hat{\mathbf{w}}_k^n) + \frac{1}{N} \sum_{n=1}^{N} \nabla_{\hat{\mathbf{w}}_k^n} F_n(\hat{\mathbf{w}}_k^n)\|_2^2$$

$$\underset{(a)}{\leq} \frac{\eta}{2} \mathbb{E} \|\nabla F(\bar{\mathbf{w}}_k)\|_2^2 + \frac{\eta}{2} \mathbb{E} \| -\frac{1}{N} \sum_{n=1}^{N} \nabla_{\mathbf{w}_k^n} F_n(\mathbf{w}_k^n) + \frac{1}{N} \sum_{n=1}^{N} \nabla_{\hat{\mathbf{w}}_k^n} F_n(\hat{\mathbf{w}}_k^n)\|_2^2$$

$$-\frac{\eta}{2}\mathbb{E}\|\nabla F(\bar{\mathbf{w}}_k)\|_2^2 - \frac{\eta}{2}\mathbb{E}\|\frac{1}{N}\sum_{n=1}^{N}\nabla_{\hat{\mathbf{w}}_k^n}F_n(\hat{\mathbf{w}}_k^n)\|_2^2 + \frac{\eta}{2}\mathbb{E}\|\nabla F(\bar{\mathbf{w}}_k) - \frac{1}{N}\sum_{n=1}^{N}\nabla_{\hat{\mathbf{w}}_k^n}F_n(\hat{\mathbf{w}}_k^n)\|_2^2$$

$$+ L_s\eta^2\mathbb{E}\|\frac{1}{N}\sum_{n=1}^{N}\nabla_{\mathbf{w}_k^n}F_n(\mathbf{w}_k^n) - \frac{1}{N}\sum_{n=1}^{N}\nabla_{\hat{\mathbf{w}}_k^n}F_n(\hat{\mathbf{w}}_k^n)\|_2^2 + L_s\eta^2\mathbb{E}\|\frac{1}{N}\sum_{n=1}^{N}\nabla_{\hat{\mathbf{w}}_k^n}F_n(\hat{\mathbf{w}}_k^n)\|_2^2$$

$$= (L_s\eta^2 + \frac{\eta}{2})\mathbb{E}\| - \frac{1}{N}\sum_{n=1}^{N}\nabla_{\mathbf{w}_k^n}F_n(\mathbf{w}_k^n) + \frac{1}{N}\sum_{n=1}^{N}\nabla_{\hat{\mathbf{w}}_k^n}F_n(\hat{\mathbf{w}}_k^n)\|_2^2$$

$$- (\frac{\eta}{2} - L_s\eta^2)\mathbb{E}\|\frac{1}{N}\sum_{n=1}^{N}\nabla_{\hat{\mathbf{w}}_k^n}F_n(\hat{\mathbf{w}}_k^n)\|_2^2 + \frac{\eta}{2}\mathbb{E}\|\nabla F(\bar{\mathbf{w}}_k) - \frac{1}{N}\sum_{n=1}^{N}\nabla_{\hat{\mathbf{w}}_k^n}F_n(\hat{\mathbf{w}}_k^n)\|_2^2$$

$$\overset{\leq}{\scriptstyle(b)} (L_s\eta^2 + \frac{\eta}{2})\frac{1}{N}\sum_{n=1}^{N}\mathbb{E}\|\nabla_{\mathbf{w}_k^n}F_n(\mathbf{w}_k^n) - \nabla_{\hat{\mathbf{w}}_k^n}F_n(\hat{\mathbf{w}}_k^n)\|_2^2$$

$$- (\frac{\eta}{2} - L_s\eta^2)\mathbb{E}\|\frac{1}{N}\sum_{n=1}^{N}\nabla_{\hat{\mathbf{w}}_k^n}F_n(\hat{\mathbf{w}}_k^n)\|_2^2 + \frac{\eta}{2}\mathbb{E}\|\nabla F(\bar{\mathbf{w}}_k) - \frac{1}{N}\sum_{n=1}^{N}\nabla_{\hat{\mathbf{w}}_k^n}F_n(\hat{\mathbf{w}}_k^n)\|_2^2, \tag{57}$$

where $(a)$ holds because $2ab \le a^2 + b^2$ and $-2ab = -a^2 - b^2 + (a-b)^2$, and $(b)$ is due to $\|\sum_{i=1}^{n}a_i\|_2^2 \le n\sum_{i=1}^{n}\|a_i\|_2^2$. Considering the distance between the gradient of the average model and the filled model, we have

$$\mathbb{E}\|\nabla F(\bar{\mathbf{w}}_k) - \frac{1}{N}\sum_{n=1}^{N}\nabla_{\hat{\mathbf{w}}_k^n}F_n(\hat{\mathbf{w}}_k^n)\|_2^2$$

$$= \mathbb{E}\|\frac{1}{N}\sum_{n=1}^{N}\nabla F(\bar{\mathbf{w}}_k) - \frac{1}{N}\sum_{n=1}^{N}\nabla_{\hat{\mathbf{w}}_k^n}F_n(\hat{\mathbf{w}}_k^n)\|_2^2$$

$$\le \frac{1}{N}\sum_{n=1}^{N}\mathbb{E}\|\nabla F(\bar{\mathbf{w}}_k) - \nabla_{\hat{\mathbf{w}}_k^n}F_n(\hat{\mathbf{w}}_k^n)\|_2^2$$

$$\overset{\leq}{\scriptstyle(a)} \frac{L_s}{N}\sum_{n=1}^{N}\mathbb{E}\|\bar{\mathbf{w}}_k - \hat{\mathbf{w}}_k^n\|_2^2$$

$$\overset{\leq}{\scriptstyle(b)} 4L_s\eta^2 E^2 G^2, \tag{58}$$

where $(a)$ follows from Assumption 1 and $(b)$ is derived by (50). Consider there are $M$ total neurons, i.e., $\sum_{l=1}^{L}m_l = M$. Based on Lemma 1, we have

$$\mathbb{E}\|\nabla_{\hat{\mathbf{w}}_k^n}f_n(\hat{\mathbf{w}}_k^n) - \nabla_{\mathbf{w}_k^n}f_n(\mathbf{w}_k^n)\|_2^2$$

$$\le G_h^2 H^2\sum_{l=1}^{L}\mathbb{E}\sum_{i\in S_{l,k}}\|\mathbf{e}_{l,i,k}\|_2^2 + \sum_{l=1}^{L}\mathbb{E}\sum_{i\in S_{l,k}^c}\|\nabla_{\hat{\mathbf{w}}_{l,i,k}^n}f_n(\hat{\mathbf{w}}_k^n)\|_2^2$$

$$\overset{\leq}{\scriptstyle(a)} G_h^2 H^2\sum_{l=1}^{L}\sum_{i\in S_{l,k}}\epsilon^2 + \alpha G^2$$

$$\le G_h^2 H^2 rM\epsilon^2 + \alpha G^2, \tag{59}$$

where $(a)$ is derived from Lemma 1 and (52).

By bringing (58) and (59) back to (57), we can obtain

$$\mathbb{E}(F(\bar{\mathbf{w}}_{k+1}) - F(\bar{\mathbf{w}}_k))$$

$$\le (L_s\eta^2 + \frac{\eta}{2})(G_h^2 H^2 rM\epsilon^2 + \alpha G^2) - (\frac{\eta}{2} - L_s\eta^2)\mathbb{E}\|\frac{1}{N}\sum_{n=1}^{N}\nabla_{\hat{\mathbf{w}}_k^n}F_n(\hat{\mathbf{w}}_k^n)\|_2^2 + 2L_s\eta^3 E^2 G^2, \tag{60}$$

Summing both sides of (60) from $k = 1$ to $K$ gets

$$\mathbb{E}(F(\bar{\mathbf{w}}_{K+1}) - F(\bar{\mathbf{w}}_1))$$

$$\le K(L_s\eta^2 + \frac{\eta}{2})(G_h^2 H^2 rM\epsilon^2 + \alpha G^2) - \sum_{k=1}^{K}(\frac{\eta}{2} - L_s\eta^2)\mathbb{E}\|\frac{1}{N}\sum_{n=1}^{N}\nabla_{\hat{\mathbf{w}}_k^n}F_n(\hat{\mathbf{w}}_k^n)\|_2^2 + 2KL_s\eta^3 E^2 G^2. \tag{61}$$

Considering $\bar{\mathbf{w}}_1 = \mathbf{w}_1$ and $F_* \leq F(\bar{\mathbf{w}}_{K+1})$, we re-organize (61) by moving $\sum_{k=1}^{K}(\frac{\eta}{2} - L_s\eta^2)\mathbb{E}\|\frac{1}{N}\sum_{n=1}^{N}\nabla_{\hat{\mathbf{w}}_k^n}F_n(\hat{\mathbf{w}}_k^n)\|_2^2$ to left hand divide both sides by $\eta$, and can obtain:

$$\sum_{k=1}^{K}(\frac{1}{2} - L_s\eta)\mathbb{E}\|\frac{1}{N}\sum_{n=1}^{N}\nabla_{\hat{\mathbf{w}}_k^n}F_n(\hat{\mathbf{w}}_k^n)\|_2^2$$

$$\leq \frac{F(\mathbf{w}_1) - F_*}{\eta} + K(L_s\eta + \frac{1}{2})(G_h^2H^2rM\epsilon^2 + \alpha G^2) + 2KL_s\eta^2E^2G^2. \tag{62}$$

Note that $\hat{\mathbf{w}}_k = \mathbf{w}_t$ when $t*E = k$, we have

$$\sum_{t=1}^{T}\mathbb{E}\|\nabla_{\mathbf{w}_k^n}F(\mathbf{w}_k^n)\|_2^2 \leq \sum_{k=1}^{K}\mathbb{E}\|\frac{1}{N}\sum_{n=1}^{N}\nabla_{\hat{\mathbf{w}}_k^n}F_n(\hat{\mathbf{w}}_k^n)\|_2^2. \tag{63}$$

Also, we set

$$\eta < \frac{1}{4L_s} \tag{64}$$

such that

$$\frac{1}{4} \leq \frac{1}{2} - L_s\eta. \tag{65}$$

Jointly considering the two above inequalities together, we can get

$$\sum_{t=1}^{T}\mathbb{E}\|\nabla_{\mathbf{w}_t}F(\mathbf{w}_t)\|_2^2$$

$$\leq \sum_{k=1}^{K}\mathbb{E}\|\frac{1}{N}\sum_{n=1}^{N}\nabla_{\hat{\mathbf{w}}_k^n}F_n(\hat{\mathbf{w}}_k^n)\|_2^2$$

$$\leq \frac{4(F(\mathbf{w}_1) - F_*)}{\eta} + 4K(L_s\eta + \frac{1}{2})(G_h^2H^2rM\epsilon^2 + \alpha G^2) + 8KL_s\eta^2E^2G^2. \tag{66}$$

By approximately considering $K = T*E$, we have

$$\frac{1}{T}\sum_{t=1}^{T}\mathbb{E}\|\nabla_{\mathbf{w}_t}F(\mathbf{w}_t)\|_2^2 \leq \frac{4(F(\mathbf{w}_1) - F_*)}{T\eta} + 4E(L_s\eta + \frac{1}{2})(G_h^2H^2rM\epsilon^2 + \alpha G^2) + 8EL_s\eta^2E^2G^2. \tag{67}$$

Setting $\eta = \sqrt{\frac{1}{T}}$, we can obtain that

$$\frac{1}{T}\sum_{t=1}^{T}\|\nabla_{\mathbf{w}_t}F(\mathbf{w}_t)\|_2^2 \leq \frac{4(F(\mathbf{w}_1) - F_*)}{\sqrt{T}} + 4E(\frac{L_s}{\sqrt{T}} + \frac{1}{2})(G_h^2H^2rM\epsilon^2 + \alpha G^2) + \frac{8EL_sE^2G^2}{T}, \tag{68}$$

which completes the proof. □

## D MORE EXPERIMENTAL DETAILS

The expermental setup for Table (1) (5(a)) (5(b)) (5(c)) Figure (10) and Figure (5) is listed in Table 3.

**Table 3: Experimental setup details on EMNIST, CIFAR-10 and CIFAR-100.**

|  |  | EMNIST | CIFAR-10 | CIFAR-100 |
|---|---|---|---|---|
| Local Epoch |  | 2 | 2 | 2 |
| Batch Size |  | 16 | 16 | 16 |
| Learning Rate |  | 0.001 | 0.001 | 0.001 |
| Decay Schedule | High Data Heterogeneity | None | None | None |
|  | Low Data Heterogeneity | None | None | None |
| Communication Rounds | High Data Heterogeneity | 1000 | 2500 | 2500 |
|  | Low Data Heterogeneity | 1000 | 2500 | 2500 |
| Optimizer |  | SGD | SGD | SGD |
| Momentum |  | 0.9 | 0.9 | 0.9 |
| Weight Decay |  | 5.00E-04 | 5.00E-04 | 5.00E-04 |
| Inference Batch |  | all | all | all |

The Impact of client model heterogeneity distribution in CIFAR-100 Figure 10

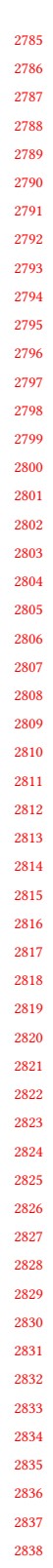

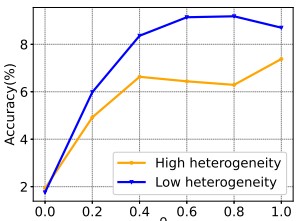

**Figure 10: Impact of client model heterogeneity distribution in CIFAR-100.**

The specify data of Figure 10 is listed in table 4.

**Table 4: Impact of client model heterogeneity distribution in CIFAR-100.**

| CIFAR-100 | $\rho$ | | | | | |
|---|---|---|---|---|---|---|
| | 0 | 0.2 | 0.4 | 0.6 | 0.8 | 1 |
| High Data Heterogeneity(%) | 1.93 | 4.92 | 6.63 | 6.44 | 6.29 | 7.38 |
| Low Data Heterogeneity(%) | 1.76 | 5.98 | 8.36 | 9.14 | 9.18 | 8.70 |

The specify data of Figure 4(a) is listed in table 5.

**Table 5: Impact of client model heterogeneity distribution in EMNIST.**

| EMNIST | $\rho$ | | | | | |
|---|---|---|---|---|---|---|
| | 0 | 0.2 | 0.4 | 0.6 | 0.8 | 1 |
| High Data Heterogeneity(%) | 23.58 | 80.09 | 84.92 | 80.43 | 88.20 | 88.53 |
| Low Data Heterogeneity(%) | 51.19 | 93.44 | 96.12 | 94.79 | 95.13 | 94.33 |

The specify data of Figure 4(b) is listed in table 6.

**Table 6: Impact of client model heterogeneity distribution in CIFAR-10.**

| CIFAR-10 | $\rho$ | | | | | |
|---|---|---|---|---|---|---|
| | 0 | 0.2 | 0.4 | 0.6 | 0.8 | 1 |
| High Data Heterogeneity(%) | 17.87 | 30.53 | 36.29 | 38.17 | 39.01 | 40.74 |
| Low Data Heterogeneity(%) | 19.54 | 37.92 | 43.31 | 47.06 | 49.86 | 53.06 |

**Resource Savage of FedDSE** The computation and communication costs are also obviously related to the consumption of energy. Besides, they are highly related to intelligent service quality in terms of timeliness. To this end, via extracting neurons and only training sub-models on the edge device, the method of our paper promotes the development of edge intelligence by reducing energy consumption, memory footprint, and computational and communication cost. Besides, the benefits of our method can be found in Table 1 as training the ResNet18, where our method can reduce three types of cost, thus also reducing the energy consumption. Thanks again for this constructive comment that amplifies the impact of our method. We will add these discussions to the refined manuscript.

Table 1. Different metrics of different methods. 1/3 of neurons are extracted from the full model. The batch size is 8.

Method Memory (MB) Computation (GFlops) Communication (MB) FedAvg 569.67 14.48 44.59 FedDSE 188.17 5.68 16.17

**Table 7: Comparison of resource consumption between FedDSE and FedAvg.**

| Method | Memory (MB) | Computation (GFlops) | Communication (MB) |
|---|---|---|---|
| FedAvg | 569.67 | 14.48 | 44.59 |
| FedDSE | 188.17 | 5.68 | 16.17 |

