# OpenReview forum: "FedDSE: Distribution-aware Sub-model Extraction for Federated Learning over Resource-constrained Devices"
_ACM.org/TheWebConf/2024/Conference — TheWebConf24_

### Official Review · Reviewer_hTfz · 2023-11-04

**Novelty:** 4
**Technical Quality:** 4

**Review:**

The paper argues for extracting sub-models from global model based on the general assumption that model trained in federated setting has to encounter heterogenous data. To tackle this paper basically proposes a personalization technique that paper call sub-model extraction. Given that there are several baselines that are missing from the paper that is making it hard to see the usefulness of the method. Simpler one would be to just fine-tune model bit more a local client, second is FedProx and related ones that deals with highly heterogenous data. Another important baseline is FedL2P [1].

[1] Lee, Royson, et al. "FedL2P: Federated Learning to Personalize." arXiv preprint arXiv:2310.02420 (2023).

**Questions:**

I also have serious doubts about Section 6 on being valuable, does this analysis applied to any model architecture, different optimizer and federated aggregation technique?

**Reviewer Confidence:**

3: The reviewer is confident but not certain that the evaluation is correct

**Scope:**

3: The work is somewhat relevant to the Web and to the track, and is of narrow interest to a sub-community

---

### Official Review · Reviewer_ZCnS · 2023-11-17

**Novelty:** 4
**Technical Quality:** 4

**Review:**

The paper proposes a sub-model extraction-based method for federated learning over resource-constrained devices. The sub-model extraction considers the statistical heterogeneity across clients by letting clients adaptively extract neurons from the model according to the activation over the local dataset.

Strength:
+ The paper is well-written, presenting clear explanations of the motivation, methodology, and evaluation results.
+ The paper provides solid theoretical analyses, demonstrating the superior performance of FedDSE.

Weakness:
- The method requires additional computation for inference and calculating the activation in order to choose neurons. It would cost high for resource-constrained devices, especially when dealing with large models like extensive language models. In addition, the server still needs to send the entire model to clients in order for sub-model extraction at local, which implies a high communication cost. I see there is Table 7 for comparing the resource consumption between FedDSE and FedAvg. Do these values include the overhead for inference and calculating the activation? In addition, FedAvg is the baseline FL that cannot work with resource-constrained devices. There is a lack of comparison in resource consumption with methods designed for resource-constrained FL, such as FedRolex.
- There are related works that do not require client-side inference for the entire large model for sub-model extraction and also consider the data heterogeneity. The comparison and discussion are missing.
[1] PerFedMask: Personalized Federated Learning with Optimized Masking Vectors. ICLR 2022.
[2] FedDUAP: Federated Learning with Dynamic Update and Adaptive Pruning Using Shared Data on the Server. IJCAI 2022.

Thanks the authors for the reply. I acknowledge I have read the rebuttal.

**Questions:**

1. How is the resource consumption of FedDSE compared to other methods for resource-constrained FL, such as FedRolex?
2. What is the advantage of FedDSE over related methods [1][2]?
3. I’d like to see how local sub-models evolve during the training. Soes the sub-model for each client reach a stable state after a certain number of training rounds?
4. What is the number of clients? It mentions 100 clients on page 8. Is this the setting for every dataset?
5. While there are two levels of data heterogeneity by setting different numbers of classes, if two clients have the same set of classes, do they have the same data distribution?

**Reviewer Confidence:**

4: The reviewer is certain that the evaluation is correct and very familiar with the relevant literature

**Scope:**

4: The work is relevant to the Web and to the track, and is of broad interest to the community

---

### Official Review · Reviewer_MGkG · 2023-11-22

**Novelty:** 6
**Technical Quality:** 6

**Review:**

This paper proposes FedDSE for federated learning over resource-constrained devices which addresses the issue of conflicts among clients with different data distributions in federated learning and introduces a novel approach to extract sub-models for each client based on their data distribution. The proposed method aims to reduce conflicts and improve training efficiency by selecting neurons with the largest activation values for each client. The mentioned insights and the corresponding theoretical analyses provide a strong assurance of the reliability of the experimental results. FedDSE outperforms SOTA baselines from the experimental results.

**Pros:**
1. The writing and presentation are easy to follow.
2. The problem of statistical heterogeneity in the current FL system is common.
3. Good insights and strong theoretical analysis.
4. Analysis and interpretation of different hyperparameters in ablation study.

**Cons:**
1. The datasets are small(EMNIST & CIFAR10/100) and not enough. Why not use the Stack Overflow dataset which is used in FedRolex?
2. Edge devices may not have enough memory to run inference for the whole global model, especially nowadays when pre-trained models are getting larger and larger, so the approach in this paper has some limitations.

**Questions:**

1. Refers to Cons (1).
2. What if the edge device can not perform inference on the global model?

**Reviewer Confidence:**

2: The reviewer is willing to defend the evaluation, but it is likely that the reviewer did not understand parts of the paper

**Scope:**

3: The work is somewhat relevant to the Web and to the track, and is of narrow interest to a sub-community

---

### Official Review · Reviewer_6oxb · 2023-11-23

**Novelty:** 4
**Technical Quality:** 4

**Review:**

The main content of this paper is to propose a new method called FedDSE for extracting sub-models according to data distribution in federated learning (FL) to solve the problem of statistical heterogeneity on resource-constrained devices. The FedDSE approach allows each client to adaptively extract neurons from the entire model based on activations on its local dataset. This approach aims to reduce conflicts between clients with different data distributions and improve classification accuracy and convergence speed.

pros:

1. Consideration of statistical heterogeneity enables clients with different data distributions to reduce conflicts and performance degradation.
2.: edDSE convergence theory was established and proved that the method has asymptotic convergence rate.

cons:

1) There is a lack of analysis and comparison of the latest sota and similar methods, such as the latest sota: DepthFL[1] and Flado[2], which also studies the sparsity of single neurons based on PT-based methods to improve the impact of non-iid on the model.
2) The author spends a lot of space introducing the limited resources of edge devices, such as limited memory, energy and communication, etc., but does not compare communication costs, Flops, etc. with other PT-based Methods.
3) This paper is designed and improved for non-iid, but experiments ar only e conducted under the  division of the number of classes each client has, and no experiments are conducted under dirichlet division.
4) In PT-based Methods such as HeteroFL and FedRolex, the common local epoch is 5. Why did this article choose 3?

[1] DEPTHFL: DEPTHWISE FEDERATED LEARNING FOR HETEROGENEOUS CLIENTS,ICLR2023
[2] Adaptive Channel Sparsity for Federated Learning under System Heterogeneity, CVPR 2023

**Questions:**

1. There are also many other methods that are used in resource-constrained federated learning, such as model sparsification and matrix decomposition. What are the differences and advantages of PT-based method compared with these methods?

**Reviewer Confidence:**

3: The reviewer is confident but not certain that the evaluation is correct

**Scope:**

3: The work is somewhat relevant to the Web and to the track, and is of narrow interest to a sub-community

---

### Official Review · Reviewer_Z6G5 · 2023-11-23

**Novelty:** 4
**Technical Quality:** 5

**Review:**

This paper presents FedDSE, a novel approach which performs distinct sub-model extractions on each device based on the activation values, addressing the challenge of limited device resources while taking into account the data heterogeneity. The extensive experiments demonstrate the effectiveness of FedDSE.

The writing of this paper is well-organised and easy to follow, except for some typos and interpretation of some experimental phenomena.

Overall, I enjoy the paper. I have a few concerns, however.

The effectiveness of FedDSE. The interpretation of model convergence with FedDSE is confusing in Section 7.2.

The generalization of FedDSE. The improvement on the CIFAR100 dataset is limited, will there be any more improvement comparing other methods on larger models and more complex datasets.

The computation efficiency of FedDSE. The additional computational overhead associated with obtaining the activation output during sub-model extraction has an unclear impact on the efficiency of local training.

**Questions:**

At line 783, page 7, the experiments on CIFAR100 dataset have shown that it became difficult to select the active neurons. Does this mean that FedDSE may fail to handle the larger neural network architecture?

At line 810, page 7, ``Figure 5 shows that the accuracy increases as 𝜌 increases on the whole.'' makes me confused.

At line 822, page 8, the interpretation of the experimental phenomena in Figure 4 is convincing. The variable is not the number of models, so how could this reach a verdict that the model convergence is related to the number of models? At the meantime, the fluctuation in Figure 4(a) under low data heterogeneity may due to the limited expressive capacity of the model. I do not agree with the conclusion ``proving that a large global model is not a prerequisite for fast convergence.''

At line 689, page 6, why is the batch normalization replaced by the static batch normalization?

The time cost. How much does FedDSE affect the computational efficiency of the learning process on local devices？

**Ethics Review Description:**

Nothing

**Reviewer Confidence:**

3: The reviewer is confident but not certain that the evaluation is correct

**Scope:**

3: The work is somewhat relevant to the Web and to the track, and is of narrow interest to a sub-community

---

### Decision · Program_Chairs · 2024-01-22

**Decision:**

Accept

**Comment:**

This paper proposes FedDSE to perform distinct sub-model extractions on each device based on the activation values and benefit federated learning across resource-constrained device clusters. The contribution is well supported by both theoretical analysis and empirical evaluations. Reviewers acknowledge this paper's technical/theoretical contributions and demonstrate no objections. Thus, the area chair recommends acceptance. The authors are encouraged to incorporate the discussions and ablation studies suggested by the reviewers in a camera-ready version, with better-highlighted novelty and motivation.